# Architecture of the RNA polymerase II-Paf1C-TFIIS transcription elongation complex

Youwei Xu[1], Carrie Bernecky[1], Chung-Tien Lee[2,3], Kerstin C. Maier[1], Björn Schwalb[1], Dimitry Tegunov[1], Jürgen M. Plitzko[4], Henning Urlaub[2,3] & Patrick Cramer[1]

The conserved polymerase-associated factor 1 complex (Paf1C) plays multiple roles in chromatin transcription and genomic regulation. Paf1C comprises the five subunits Paf1, Leo1, Ctr9, Cdc73 and Rtf1, and binds to the RNA polymerase II (Pol II) transcription elongation complex (EC). Here we report the reconstitution of Paf1C from *Saccharomyces cerevisiae*, and a structural analysis of Paf1C bound to a Pol II EC containing the elongation factor TFIIS. Cryo-electron microscopy and crosslinking data reveal that Paf1C is highly mobile and extends over the outer Pol II surface from the Rpb2 to the Rpb3 subunit. The Paf1-Leo1 heterodimer and Cdc73 form opposite ends of Paf1C, whereas Ctr9 bridges between them. Consistent with the structural observations, the initiation factor TFIIF impairs Paf1C binding to Pol II, whereas the elongation factor TFIIS enhances it. We further show that Paf1C is globally required for normal mRNA transcription in yeast. These results provide a three-dimensional framework for further analysis of Paf1C function in transcription through chromatin.

[1] Department of Molecular Biology, Max-Planck-Institute for Biophysical Chemistry, Max Planck Society, Am Fassberg 11, Göttingen 37077, Germany. [2] Bioanalytical Mass Spectrometry, Max-Planck-Institute for Biophysical Chemistry, Am Fassberg 11, Göttingen 37077, Germany. [3] Bioanalytics Group, Institute for Clinical Chemistry, University Medical Center, Göttingen, Robert-Koch-Strasse 40, Göttingen 37075, Germany. [4] Department of Molecular Structural Biology, Max-Planck-Institute for Biochemistry, Am Klopferspitz 18, Martinsried 82152, Germany. Correspondence and requests for materials should be addressed to P.C. (email: pcramer@mpibpc.mpg.de).

The polymerase-associated factor 1 (Paf1) complex (Paf1C) is a general and conserved RNA polymerase II (Pol II) transcription elongation factor[1]. Paf1C was first identified through its copurification with Pol II from yeast cells[2,3]. Yeast Paf1C comprises the subunits Paf1, Leo1, Ctr9, Cdc73 and Rtf1 (refs 1,4). Paf1C shows genetic interactions with the yeast transcription elongation factors Spt4-Spt5 and Spt16-Pob3, the counterparts of human DSIF and FACT, respectively[5]. Paf1C also associates with transcribed regions *in vivo*[6], suggesting that it is a transcription elongation factor. Paf1C subunits are required for efficient transcription *in vivo*[7].

Paf1C has multiple roles in chromatin transcription. Yeast Paf1C functions in methylation of histone H3 by Set1 and Dot1, thus linking transcription elongation to chromatin methylation[8]. In yeast, Rtf1 binds the chromatin remodeller Chd1 (ref. 9) and is required for ubiquitination of histone H2B (refs 10–12) and histone methylation[13]. *Drosophila* Rtf1 also functions in histone methylation, gene expression and Notch signalling[14]. Human Paf1C binds to histone H3 tails with dimethylated histone H3 arginine17 (ref. 15).

Paf1C also has important functions that are not directly related to chromatin. Paf1C is required for cotranscriptional RNA 3′-processing[16,17]. Human Cdc73 physically interacts with protein complexes required for 3′-processing[18]. Paf1C also represses cryptic transcription[19] and is implicated in cellular differentiation[20] and human cancer[21–23]. Paf1C represses gene silencing by small RNAs in *Schizosaccharomyces pombe*[24], and

Leo1 is involved in heterochromatin spreading[25]. Paf1C has recently been found to regulate Pol II phosphorylation, promoter–proximal pausing and release into gene bodies[26,27]. Paf1C is also involved in the resolution of transcription–replication conflicts[28].

Paf1C is generally recruited to transcribed units, apparently entering the Pol II elongation complex (EC) downstream of the transcription start site, and exiting at the polyadenylation (pA) site[29]. There is evidence that Paf1C recruitment to Pol II requires direct contacts with Pol II and additional contacts with Pol II-associated factors. Paf1C recruitment *in vivo* requires the Bur1-Bur2 kinase[30], and is aided by Spt4 (ref. 31). Paf1C and its Cdc73 subunit bind the phosphorylated C-terminal domain of Pol II and the phosphorylated C-terminal repeat region (CTR) of Spt5, which is also a general elongation factor[32]. A Plus3 domain in Rtf1 can bind the Spt5 CTR[33]. Rtf1 is however not stably associated with Paf1C in all species and is not required for Paf1C recruitment in human cells, where it has non-overlapping functions[34]. In fission yeast, Rtf1 also has other functions[35]. The C-terminal GTPase-like domain of Cdc73 is important for chromatin association of Paf1C (ref. 36). Leo1 is also important for Paf1C recruitment and binds RNA[37].

Structural studies have revealed that Paf1C is a modular and flexible complex with several structured regions. The crystal structure of a complex of regions in Paf1 and Leo1 revealed antiparallel β-sheets for heterodimerization[38]. This study also showed that Ctr9 is a scaffold for Paf1C onto which the Paf1-

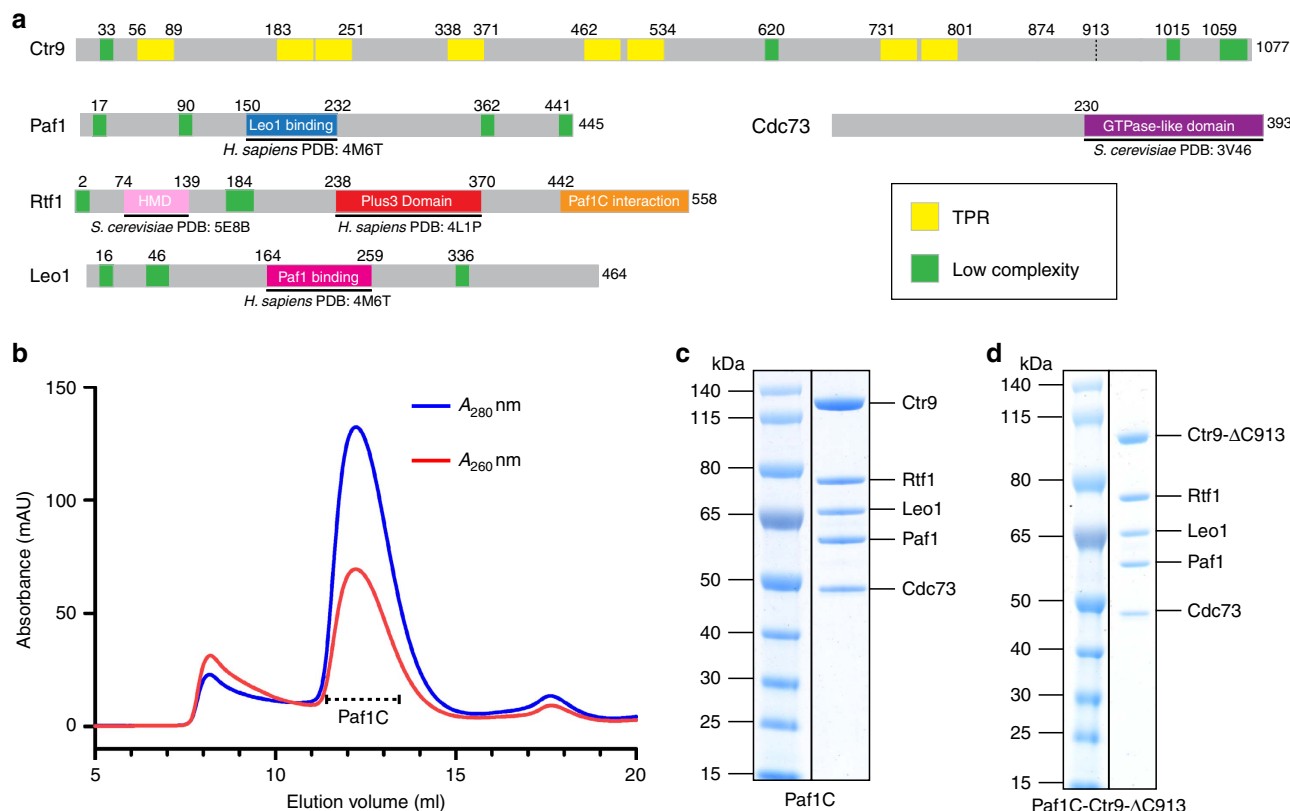

**Figure 1 | Preparation of recombinant Paf1C.** (**a**) Predicted domain structure of the five Paf1C subunits from *S. cerevisiae*. The predicted tetratrico peptide repeat motifs (yellow) and low complexity regions (green) were defined using SMART[42]. Structurally resolved regions are indicated with black lines. The Plus3 domain in Rtf1 (red) binds single-stranded DNA[40] and the Spt5 CTR[41]. A Paf1C-interacting region (orange) in Rtf1 (ref. 13) was confirmed in this work. Histone modification domain (HMD, pink) in Rtf1 directly contacts ubiquitin conjugase Rad6 *in vivo*[44]. (**b**) Size-exclusion chromatogram (Superose 6 10/300; GE Healthcare) of recombinant full-length Paf1C indicates a pure complex free of nucleic acids. (**c**) Coomassie-stained SDS–PAGE analysis of recombinant five-subunit Paf1C after size-exclusion chromatography reveals the presence of all five subunits in apparently stoichiometric amounts. Molecular weight markers are indicated on the left. The identity of subunits was confirmed by mass spectrometry. (**d**) Same as **c** but for purified complex Paf1C-Ctr9-ΔC913.

Leo1 heterodimer and Cdc73 assemble. The structure of the GTPase-like domain in the C-terminal region of yeast Cdc73 was also solved[36,39]. In addition, structures were reported for the Plus3 domain of human Rtf1 (refs 40,41), and for the Plus3 domain in complex with a phosphorylated Spt5 CTR repeat[41]. There is however no structural information on Ctr9, the largest Paf1C subunit.

Despite its critical role in chromatin transcription and transcription-coupled events, neither the structure of Paf1C nor its location on the Pol II EC are known. Here we reconstitute yeast Paf1C from recombinant subunits, assemble a complex of Paf1C with the Pol II-TFIIS EC and locate Paf1C on the polymerase surface with a combination of electron microscopy (EM) and crosslinking. From this work emerges a trilobal architecture of Paf1C and the location of Paf1C on the elongating polymerase, providing a basis for a mechanistic analysis of transcription elongation through chromatin.

## Results

**Recombinant Paf1C.** Paf1C from the yeast *Saccharomyces cerevisiae* (*S. cerevisiae*) has a molecular weight of 340 kDa and consists of five subunits, Paf1, Leo1, Ctr9, Cdc73 and Rtf1 (Fig. 1a). Analysis of the primary and secondary structures of these subunits[42,43] predicted known structured domains[36,38–41,44] and eight tetratrico peptide repeats in Ctr9 (Fig. 1a). In addition, multiple regions of low sequence complexity were detected in all Paf1C subunits except Cdc73, consistent with the known flexibility of Paf1C (Supplementary Fig. 1a). To study Paf1C structurally, we established preparation of pure recombinant Paf1C after coexpression of its subunits in *Escherichia coli* (*E. coli*) (see Fig. 1b,c and Methods section). The five Paf1C subunits were coexpressed from three vectors in *E. coli* and the complex purified using chromatographic methods (see Fig. 1b and Methods section). We obtained ∼0.4 mg of pure Paf1C per liter of *E. coli* cell culture. Purified Paf1C contained all five subunits in apparently stoichiometric amounts (Fig. 1c).

**Paf1C structural core and flexible periphery.** To map a stable structural core of Paf1C, we identified a Ctr9 variant that lacked 164 amino acids at its C terminus (Ctr9-ΔC913), but was sufficient to form a stable complex with the other subunits after coexpression (Paf1C-Ctr9-ΔC913) (Fig. 1d). We also obtained a recombinant Paf1C variant that additionally lacked subunit Rtf1 (Paf1C-Ctr9-ΔC913-ΔRtf1; Supplementary Fig. 1b). To further delineate the core of Paf1C, we used limited proteolysis and Edman sequencing (see Supplementary Fig. 1a and Methods section) and designed deletion mutants of the remaining three subunits. The limited proteolysis also showed that Leo1 and Paf1 are unstable. Combining iterative truncations of Leo1, Paf1 and Ctr9-ΔC-913, coexpression and copurification, we defined a structural core of Paf1C that additionally lacked the C-terminal region of Paf1 and both terminal regions of Leo1.

The resulting Paf1C core contained Ctr9-ΔC913, Paf1-ΔC361, Leo1-ΔN118-ΔC376 and Cdc73 (Supplementary Fig. 1c). The defined Paf1C core comprises 1,926 amino-acid residues out of a total of 2,937 residues, that is, 65% of the total protein mass. Thus, one-third of Paf1C forms flexible regions on the periphery of the complex. The C-terminal region of Rtf1 (Rtf1-ΔN441) can interact with the Paf1C core, indicating that the C-terminal region of Paf1 and both terminal regions of Leo1 are not crucial for Rtf1 binding (Supplementary Fig. 1d). In addition, we expressed and purified the Ctr9-Paf1-Leo1 trimer (Supplementary Fig. 1e). Taken together, these results show that Paf1C contains a structured core and several flexible regions around its periphery.

**TFIIS enhances Pol II-Paf1C binding.** To investigate whether purified Paf1C binds to yeast Pol II *in vitro*, we performed pull-down assays using biotinylated Pol II coupled to streptavidin beads (see Methods section). Paf1C interacted with Pol II, albeit in a substoichiometric manner (Supplementary Fig. 2a, lane 4). Considering the known interaction of human Paf1C with TFIIS[45], we tested whether TFIIS enhances Pol II-Paf1C binding. Indeed, TFIIS strongly enhanced binding of Paf1C to Pol II in our assay (Supplementary Fig. 2a, lane 5). In contrast, an N-terminal deletion variant of TFIIS lacking 130 residues (TFIIS-ΔN130) could not enhance Pol II-Paf1C interaction (Supplementary Fig. 2b, lanes 7 and 8). These results suggest that the N-terminal domain I of TFIIS, which is mobile in the Pol II-TFIIS structure[46], interacts with Paf1C to increase its affinity to Pol II. In addition, Paf1C lacking the C-terminal domain of Ctr9 and Rtf1 retained Pol II binding (Supplementary Fig. 2c, lanes 6 and 8) and TFIIS-enhanced binding (Supplementary Fig. 2c, lanes 7 and 9). These results show that recombinant Paf1C interacts with Pol II *in vitro*, that this interaction does not require Rtf1 and that it is strongly enhanced by TFIIS (Supplementary Fig. 2d).

**Cryo-EM analysis.** To determine the structure of Paf1C bound to a Pol II EC, we prepared a complex containing the complete 12-subunit Pol II[47], a DNA-RNA scaffold[48], full-length TFIIS carrying two point mutations that render it inactive in RNA cleavage stimulation[46,49] and recombinant Paf1C lacking the Ctr9 C terminus (Paf1C-Ctr9-ΔC913) (see Methods section), which is more stable than Paf1C with full-length Ctr9. The complex contained all 18 polypeptides in apparently stoichiometric amounts after sucrose gradient ultracentrifugation (Fig. 2a). After gradient fixation[50] (Supplementary Fig. 3a), the sample contained single particles, as revealed by EM in negative stain (Supplementary Fig. 3b), and was subjected to cryo-EM data collection (see Methods section). A total of 2,741 micrographs were collected on a Titan Krios equipped with a Gatan K2 direct electron detection device (Supplementary Fig. 3c).

From the cryo-EM micrographs, a total of 947,597 particles were extracted. Particle images were processed and subjected to reference-free two-dimensional (2D) classification in RELION[51], yielding 370,841 particles after clearance. In several 2D classes, smeared densities for the peripheral Paf1C and the Pol II stalk subcomplex Rpb4-Rpb7 were observed on the polymerase surface (Fig. 2b). As a reference, we used a reconstruction of bovine Pol II[48] filtered to low resolution (50 Å). After particle polishing in RELION, three-dimensional (3D) classification was used to separate out 158,422 particles that contained Pol II-TFIIS EC that lost Paf1C.

We refined the remaining 212,419 Paf1C containing particles, which led to a structure at 5.5 Å global resolution that contained the Pol II-TFIIS EC that closely resembled the crystal structure (Supplementary Fig. 4). Although the polymerase, TFIIS and the nucleic acids are very well defined, the local resolution of different parts of Paf1C is lower, ∼10–20 Å, indicating flexibility. A globally filtered 18 Å resolution reconstruction of the Pol II-Paf1C-TFIIS EC complex shows the location of Paf1C on the Pol II-TFIIS EC (Fig. 2c). To improve Paf1C density, we performed sub-3D classification. This resulted in three different Pol II-Paf1C-TFIIS EC reconstructions that we refer to as A, B and C (see Supplementary Fig. 4 and Methods section). These reconstructions showed average resolutions of 5.7, 5.9 and 6.2 Å, respectively (Fourier shell correlation (FSC) = 0.143; see Supplementary Fig. 5a,b and Methods section) and enabled rigid-body fitting of the Pol II-TFIIS crystal structure[49,52] and the DNA-RNA model[48] (see Supplementary Fig. 5c,d and Methods section). The reconstructions did not reveal structural changes within the Pol II-TFIIS complex compared to the known crystal structure.

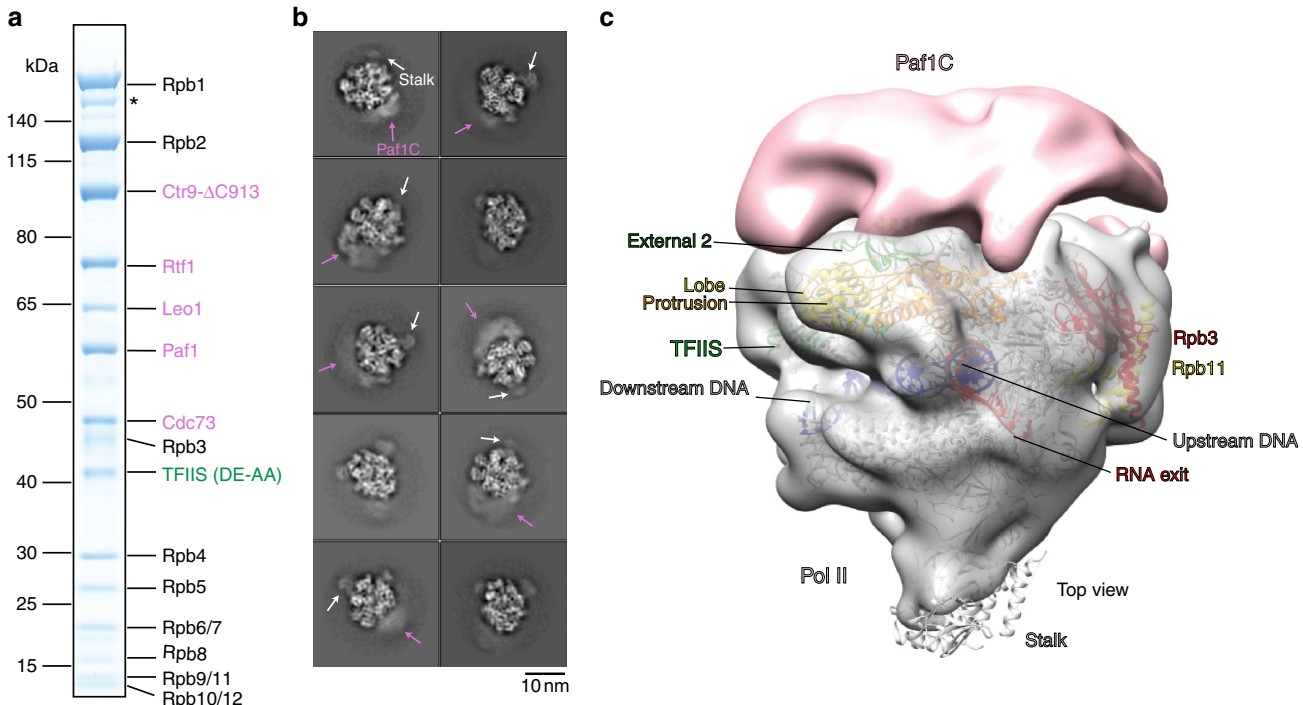

**Figure 2 | Reconstitution and cryo-EM analysis of Pol II-Paf1C-TFIIS EC.** (**a**) SDS–PAGE analysis (Coomassie staining) of Pol II-Paf1C-TFIIS EC after sucrose gradient ultracentrifugation. Subunits from Pol II and Paf1C are labelled in black and pink, respectively. The inactive variant of TFIIS is labelled in green. The asterisk marks a degradation product of Rpb1. (**b**) Ten representative reference-free 2D cryo-EM class averages reveal the known flexibility of the Pol II stalk subcomplex Rpb4/7 (white arrow) and density for Paf1C (pink arrow). Scale bar, 10 nm. (**c**) Location of Paf1C on the Pol II-TFIIS EC. A reconstruction of Pol II-Paf1C-TFIIS EC was refined from 212,419 particles after the first 3D classification and globally filtered to 18 Å resolution. Density for the Pol II-TFIIS EC subcomplex is shown in grey. Continuous Paf1C density at the outer surface of Pol II is shown in pink. The structure of the Pol II-TFIIS EC is fitted and shown in ribbon representation (non-template DNA, cyan; template DNA, blue; RNA, red, TFIIS, green; Rpb2 lobe, yellow; Rpb2 protrusion, orange; Rpb2 external 2, green; Rpb3, red; Rpb11, yellow).

**Architecture of Pol II-Paf1C-TFIIS EC.** Reconstructions A, B and C revealed different portions of Paf1C on the Pol II surface that we refer to as Paf1C parts A, B and C, respectively (Fig. 3a). The observation that these three parts of Paf1C could only be observed in different cryo-EM reconstructions shows that their relative orientation is flexible. Owing to this mobility on the Pol II surface, the local resolution of Paf1C was low (Supplementary Fig. 5e–g). When contoured at a level where no noise peaks are observed (Fig. 3b), the volumes of parts A, B and C could account for ∼50, 45 and 90 kDa of folded protein (see Methods section). Because part C overlaps with part A by ∼10 kDa, a total of ∼175 kDa of Paf1C is visible by cryo-EM, corresponding to ∼50% of the total mass of Paf1C, suggesting that most of the structured core (65%) is revealed (Supplementary Fig. 1). We combined reconstructions A–C into a composite map that is further interpreted below.

Paf1C binds to the outer surface of Pol II on the Rpb2 side, spanning from the Rpb2 external 2 and protrusion domains to subunit Rpb3 and reaching near the rim of the funnel opposite the active centre cleft (Fig. 3a,b). Paf1C part A contacts mainly the external 2 domain and likely the protrusion domain of Rpb2. Part B contacts the protrusion and part C contacts Rpb11 and Rpb3 around its helix α3. Part B also bridges between parts A and C, which otherwise do not contact each other. Taken together, cryo-EM revealed that the Paf1C core may be divided into three parts, with parts A and C contacting Pol II at the Rpb2 lobe and external 2 domains and Rpb3, respectively, and part B being highly mobile and bridging between parts A and C.

**Crosslinking analysis.** To confirm the contacts of Paf1C with Pol II, and to assign Paf1C subunits to the three parts of Paf1C, we subjected the Pol II-Paf1C-TFIIS EC to chemical crosslinking coupled to mass spectrometry (XL-MS; see Methods section). We used the crosslinking reagent bis(sulfosuccinimidyl) suberate (BS3), which reacts with lysine side chains and N termini. In each crosslinking data set, two replicates were measured. All the spectra of the crosslinks were filtered at a false discovery rate cutoff of 1% and the maximum score value (negative logarithm of E-value) >5. Each crosslink was required to have a minimal spectral count of 2 in the two replicates of each data set (see Methods section). We obtained 239 unique intersubunit cross-links (Fig. 4a, Table 1 and Supplementary Data 1). We first mapped crosslinks between Pol II subunits and between Pol II and TFIIS onto the Pol II-TFIIS EC crystal structure[49]. The Cα–Cα distances between crosslinked residues were within the allowed distances of 30 Å for 35 out of 38 intersubunit pairs. The other three crosslinks that showed longer Cα–Cα distances fell in regions with structural flexibility and higher crystallographic B-factors (Fig. 4b,c and Supplementary Data 1). In addition, we observed that Leo1 was crosslinked to two lysine residues (K78, K80) in domain I of TFIIS, consistent with our pull-down assays (see above). These results provided an internal, positive control for our crosslinking approach, and demonstrated the proximity of Paf1C to TFIIS.

**Paf1C subcomplex architecture.** The data also contained 136 intersubunit crosslinks within Paf1C (Fig. 4a, Table 1 and Supplementary Data 1). These included 25 crosslinks between Paf1 and Leo1, consistent with the known dimerization of Paf1 and Leo1 subunits. Crosslinks of the Leo1 residue K225 to Paf1 residues K167 and K170 can be rationalized with the Paf1-Leo1

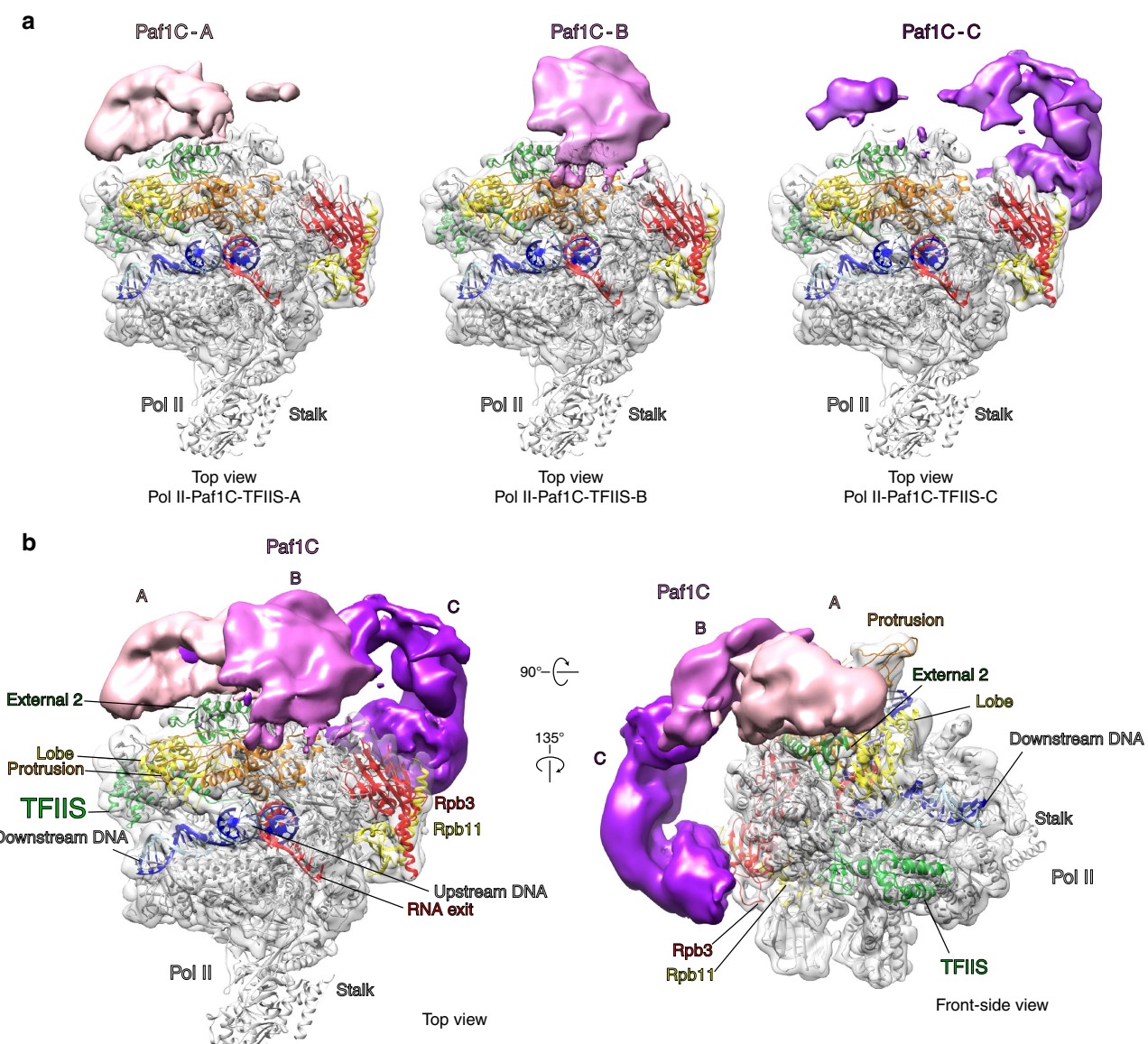

**Figure 3 | Cryo-EM structure of Pol II-Paf1C-TFIIS EC.** (**a**) Three reconstructions of Pol II-Paf1C-TFIIS EC after sub-3D classification. Density for the Pol II-TFIIS EC subcomplex is shown as a silver semitransparent surface. The electron densities for Paf1C are shown as a solid surface coloured in pink, magenta and purple for Paf1C parts A, B and C, respectively. All three maps are unsharpened and filtered according to local resolution. The fitted structure of the Pol II-TFIIS EC is shown in ribbon representation (non-template DNA, cyan; template DNA, blue; RNA, red, TFIIS, green; Rpb2 lobe, yellow; Rpb2 protrusion, orange; Rpb2 external 2, green; Rpb3, red; Rpb11, yellow). (**b**) Two views of the composited EM density for the Pol II-Paf1C-TFIIS EC from three reconstructions in **a** and the fitted structure of the Pol II-TFIIS EC. These densities of locally filtered Pol II and parts A–C are shown at the same threshold. The view on the left is from the top[80], and the view on the right is related by two rotations as indicated. The same threshold level in chimera was used to contour densities.

X-ray structure. A total of 27 crosslinks were observed between the Paf1-Leo1 dimer and Ctr9, revealing their proximity. The subassembly Paf1-Leo1-Ctr9 crosslinked to the C-terminal regions of Cdc73 and Rtf1. Consistent with this, the C-terminal region of Rtf1 suffices to bind Paf1C, because its coexpression with the Paf1C core resulted in a stable complex (Supplementary Fig. 1c). We also crosslinked free Paf1C and the obtained crosslinking pattern was similar (Supplementary Data 1).

We also obtained 49 intersubunit crosslinks between Paf1C and Pol II (Fig. 4a, Table 1, Supplementary Fig. 6 and Supplementary Data 1). Forty-four of these crosslinks map to the Rpb2 lobe, protrusion and external 2 domains, and subunits Rpb3 and Rpb11 (Supplementary Data 1), consistent with the location of Paf1C observed by cryo-EM. The lysine-rich region (K298–K369) of Leo1

gives rise to many crosslinks to the protrusion, lobe and external 2 of Rpb2, which together with extensive crosslinking between the Paf1-Leo1 dimer and the Pol II protrusion and lobe show that the Paf1-Leo1 heterodimer resides in part A. Crosslinks between Cdc73 and Pol II subunits Rpb3 and Rpb11 reveal that Cdc73 resides in part C. Indeed, part C contains a globular density that contacts Rpb3 and corresponds in size to the crystal structure of the C-terminal GTPase-like domain of Cdc73 (ref. 36) (Fig. 4d). We could fit this crystal structure to the globular density[53] such that the crosslinks between Cdc73 and Pol II were explained. Three crosslinks occurred between the fitted structure and Rpb3, whereas Cdc73 residues in the adjacent N-terminal tail crosslinked to the nearby Rpb11 subunit (Fig. 4d). Finally, Ctr9 can be assigned to part B because it crosslinks to both Paf1-Leo1 in

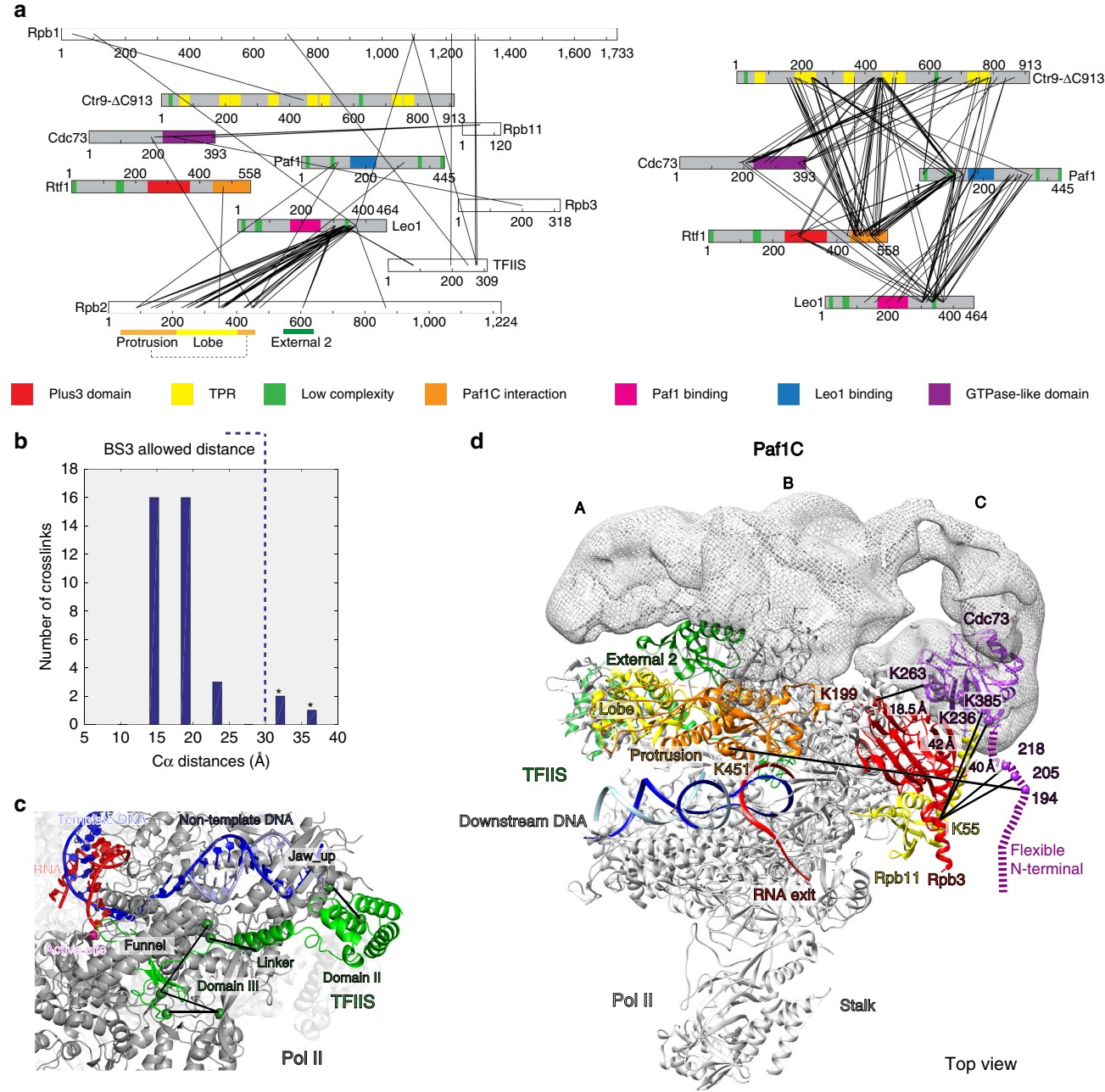

**Figure 4 | Crosslinking analysis of Pol II-Paf1C-TFIIS EC. (a)** Network diagrams of pairwise crosslinks (black lines) obtained after incubation of Pol II-Paf1C-TFIIS EC with BS3. The left diagram depicts intersubunit crosslinks between Pol II and Paf1C, Pol II and TFIIS, and between Paf1C and TFIIS. The right diagram depicts crosslinks between Paf1C subunits. Crosslinks between Pol II subunits were excluded for clarity. Paf1C subunits coloured as in Fig. 1. In addition, the regions of protrusion, lobe and external 2 in Rpb2 are indicated with orange, yellow and green bars below Rpb2 diagram, respectively. **(b)** Cα–Cα distance distribution of observed Pol II-Pol II and Pol II-TFIIS crosslinks for lysine residues that were resolved in the Pol II-TFIIS crystal structure. These crosslinks serve as a positive control for our crosslinking data. Asterisks indicate crosslinks to flexible protein regions. **(c)** Crosslinks between Pol II and TFIIS (black lines) are consistent with the known Pol II-TFIIS complex crystal structure. Colour coding as in Fig. 3. **(d)** Position of the Cdc73 C-terminal GTPase-like domain (purple) contacting Rpb3. The crystal structure of the GTPase-like domain was docked into density of the correct size and oriented according to crosslinks to Rpb3 and Rpb11 (black lines). Paf1C density is shown in mesh in the same orientation as in Fig. 3. View is from the top. The distances between site of crosslinking in Cdc73 and Rpb3/Rpb11 are revealed. The distance between K263 of Cdc73 and K199 of Rpb1 in this fit is 18.5 Å. the distances between K236 and K385 in the Cdc73 GTPase-like domain structure and K55 of Rpb11 are around 40 Å due to the flexibility of both terminal regions of the GTPase-like domain.

part A and Cdc73 in part C. Thus, Ctr9 bridges between Paf1-Leo1 and Cdc73, which both contact Pol II. Ctr9 did not crosslink efficiently to Pol II, consistent with the bridging density B that forms only limited contacts with the Pol II surface.

Combining the crosslinking data with the cryo-EM results and our mapping of flexible regions in Paf1C enabled us to derive the overall architecture of Paf1C bound to Pol II. The sizes of the EM densities can be reconciled with the Paf1C subunit molecular weights as follows. Part A reveals only about half of the Paf1-Leo1 heterodimer (∼50 out of 105 kDa), the other half is flexible. Part B reflects about half of the mass (∼45 kDa) of Ctr9-ΔC913 (105 kDa), whereas the other half of Ctr9 resides in part C, which

**Table 1 | Summary of BS3 crosslinks in Pol II-Paf1C-TFIIS EC.**

| Protein 1 | Protein 2 | Unique crosslinks BS3 |
|---|---|---|
| Paf1C | Pol II | 49 |
| Ctr9 | Rpb1 | 1 |
| | Rpb3 | 1 |
| Rtf1 | Rpb2 | 1 |
| Leo1 | Rpb1 | 2 |
| | Rpb2 | 35 |
| Paf1 | Rpb2 | 3 |
| Cdc73 | Rpb11 | 4 |
| | Rpb2 | 1 |
| | Rpb3 | 1 |
| TFIIS | Paf1C | 2 |
| | Leo1 | 2 |
| Paf1C | Paf1C | 136 |
| Ctr9 | Rtf1 | 36 |
| | Leo1 | 7 |
| | Paf1 | 20 |
| | Cdc73 | 19 |
| Rtf1 | Leo1 | 6 |
| | Paf1 | 19 |
| Leo1 | Paf1 | 25 |
| | Cdc73 | 4 |
| Pol II | Pol II | 47 |
| Pol II | TFIIS | 5 |
| | Total | 239 |

BS3, bis(sulfosuccinimidyl) suberate; EC, elongation complex; Paf1C, polymerase-associated factor 1 complex; Pol II, polymerase II.
The number of interprotein crosslinks are given. Interprotein crosslinks between Pol II subunits are not shown here for clarity. Each unique crosslink is identified by more than one crosslink. BS3 is a crosslinker with 11.4 Å spacer arm and reacts efficiently with amino groups (lysine and N terminus).

is much larger than the Cdc73 GTPase-like domain alone (∼20 kDa). Finally, Rtf1 (65 kDa) remains flexible.

**Competitive Pol II binding of Paf1C and TFIIF.** Superposition of the Pol II-Paf1C-TFIIS EC onto our previously reported Pol II initiation complex structure[54] reveals a clash between Paf1C part A and the dimerization domain of the initiation factor TFIIF bound to the outer lobe of Pol II at an overlapping position (Fig. 5a). This suggested that Paf1C and TFIIF may bind to Pol II in a competitive manner. To investigate this, we performed binding assays using analytical sucrose gradient ultracentrifugation (see Methods section). We incubated preformed Pol II-Paf1C-TFIIS complex with a 1.8-fold molar excess of TFIIF and separated the resulting complexes on a 10–30% sucrose gradient. Based on subsequent SDS–polyacrylamide gel electrophoresis (SDS–PAGE) analysis, we estimate that TFIIF replaced Paf1C in about half of the Pol II complexes (Fig. 5b). When we instead incubated preformed Pol II-TFIIF complex with Paf1C and TFIIS, some TFIIF was displaced, TFIIS was able to join the complex and Paf1C bound to a low extent (Fig. 5c). Analysis of the relevant gradient fractions by native PAGE shows that TFIIF can almost completely compete off the Pol II-bound Paf1C, but not the other way around (Fig. 5d,e). These results indicate that Paf1C and TFIIF bind to Pol II in a competitive manner, consistent with the cryo-EM reconstruction.

**Paf1C is globally required for Pol II transcription.** Thus far, it had not been demonstrated whether Paf1C is a general transcription factor in yeast. To investigate whether Paf1C is globally required for transcription *in vivo*, or whether it has gene-specific functions, we monitored RNA synthesis with 4-thiouracil-Seq (4tU-Seq) in yeast[55–57]. This method uses metabolic RNA labelling with 4tU coupled to strand-specific sequencing of labelled, newly synthesized RNA. We used 4tU-Seq to monitor RNA synthesis in strains lacking either Paf1 (paf1Δ) or Rtf1 (rtf1Δ), and compared this to a wild-type strain using global normalization based on spike-in probes[58]. Two biological replicates were measured. We found that knockout of Paf1 or Rtf1 led to a strong, global decrease in synthesis of mRNA transcripts (Fig. 6a), showing that Paf1C is globally required for normal Pol II transcription. Furthermore, the analysis of significantly downregulated transcripts in paf1Δ and rtf1Δ strains with a Venn diagram shows strong overlap between paf1Δ and rtf1Δ strains, but that the rtf1Δ strain shows 35% more downregulated RNAs, consistent with the idea that Rtf1 has Paf1C-independent roles[34] (Fig. 6b).

**Discussion**
Here, we use a combination of cryo-EM and crosslinking to show that Paf1C forms a tripartite architecture. Parts A and C contain Paf1-Leo1 and Cdc73, respectively, and contact Pol II near the Rpb2 external 2 domain and Rpb3, respectively. Ctr9 forms part B and extends into part C, forming a flexible bridge between parts A and C that is not directly contacting Pol II. These results explain published data. Ctr9 has been predicted from biochemical data to form a scaffold that bridges between Paf1-Leo1 and Cdc73 (ref. 38). The Cdc73 GTPase-like domain binds Pol II, consistent with the requirement for this domain to recruit Paf1C to chromatin[36]. The C-terminal region of Cdc73 (201–393 amino acids) has been reported to bind the Pol II C-terminal domain[35], but this region could not be localized here. Most of Rtf1 is flexible, including the Plus3 domain that binds the phosphorylated CTR of Spt5 (ref. 41), which is also flexibly connected to the EC. Our results also explain how Paf1C and TFIIS can cooperatively bind to Pol II[45]. Pull-down assays show that TFIIS facilitates Pol II-Paf1C binding via its domain I. Correspondingly, we observe crosslinking between Leo1 and TFIIS domain I, and this interaction occurs near the jaw-lobe module of Pol II. We arrive at an overall architecture of the Paf1C on the Pol II EC that explains known interactions (Fig. 7).

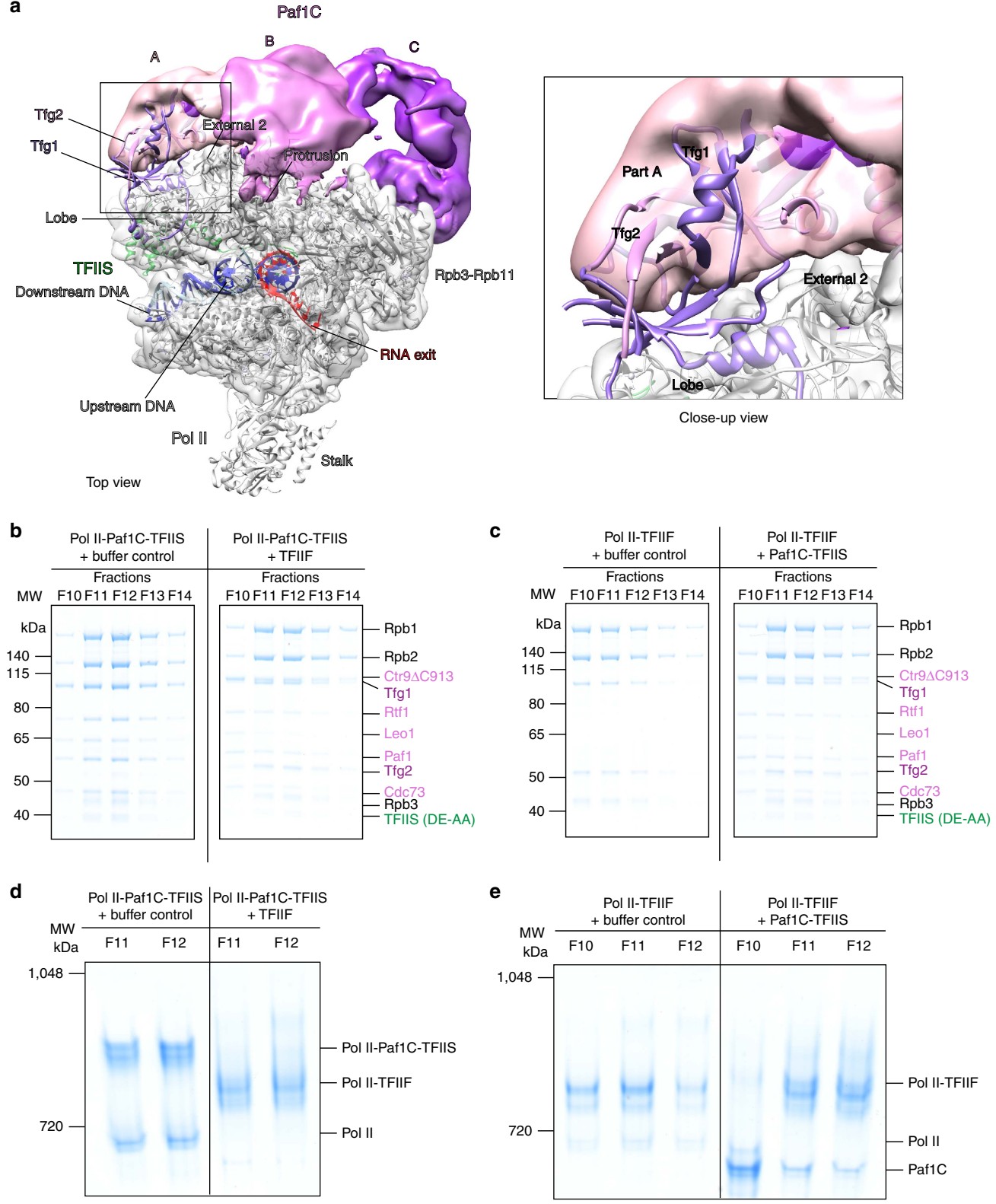

**Figure 5 | Paf1C and TFIIF compete for Pol II binding.** (**a**) Positioning the initiation factor TFIIF dimerization domain as in the Pol II initiation complex (PDB accession: 5FYW)[54] onto the Pol II-Paf1C-TFIIS EC cryo-EM reconstruction results in a clash of TFIIF with Paf1C (top view). Colours as in Fig. 3. The TFIIF dimerization domain is coloured in medium purple (TFIIF subunit Tfg1) and plum (Tfg2). A close-up view of the clashing region is shown on the right. (**b,c**) SDS–PAGE analysis of TFIIF and Paf1C competitive binding to Pol II after analytical sucrose gradient ultracentrifugation. In each panel, the gels on the left depict complexes without competitor protein. The gels on the right represent complexes formed after competitor addition. In **b**, a 1.8-fold molar excess of TFIIF was added to preformed Pol II-Paf1C-TFIIS EC. In **c**, Paf1C-TFIIS was added to preformed Pol II-TFIIF complex. (**d,e**) Native PAGE analysis of indicated fractions from the analysis in **b**. In our minimal *in vitro* system, TFIIF is able to largely compete out Paf1C. The identity of the proteins in the indicated complexes was confirmed by mass spectrometry.

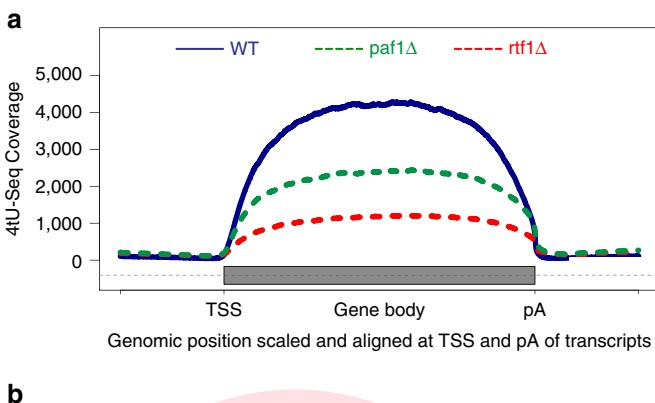

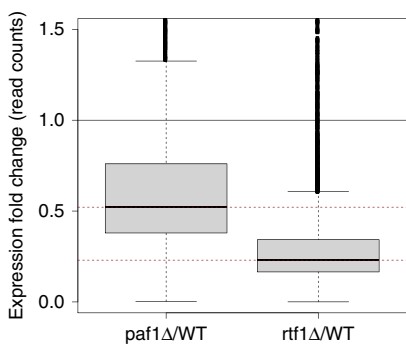

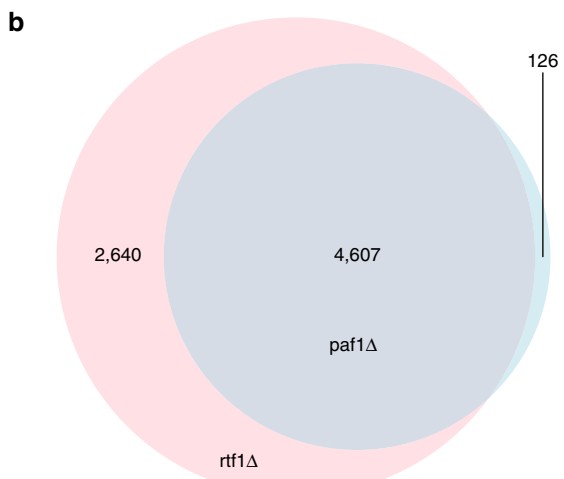

**Figure 6 | Paf1C is globally required for Pol II transcription in yeast.** (**a**) Left panel: coverage of newly synthesized RNA measured by 4tU-Seq in WT (solid blue line), rtf1Δ (dashed red line) and paf1Δ yeast cells (dashed green line). 4tU-Seq signals were globally normalized using spike-in probes. The gene body is defined as the region spanning from the transcription start site (TSS) to the pA site and is depicted as a grey box. Right panel: box plot showing the expression fold change of read counts for paf1Δ and rtf1Δ versus the wild-type (WT) strain. Dashed lines indicate median fold (0.52 for paf1Δ and 0.23 for rtf1Δ). The majority of mRNAs have signals below 1 (indicated by the solid black line). mRNA transcripts that are downregulated make up 85% for paf1Δ and 94% for rtf1Δ of all mRNAs with a coverage of at least 2 in the WT strain. (**b**) Analysis of significantly downregulated transcripts in paf1Δ and rtf1Δ strains using a Venn diagram. Most of the downregulated transcripts in paf1Δ (4,607 out of 4,733) strain are also affected by the rtf1Δ mutation, although 2,640 transcripts are uniquely downregulated in the rtf1Δ strain.

We further show that Paf1C is generally required for transcription *in vivo*, and thus our results elucidate aspects of the general transcription cycle. In particular, they provide insights into the transition from transcription initiation to elongation, when initiation factors are replaced by elongation factors on the Pol II surface[29,59,60]. We suggest that in the initiation complex, Paf1C cannot bind Pol II because the site for binding part A is occupied by TFIIF, which binds the lobe. Similarly, TFIIE blocks the site for binding the elongation factor Spt4-Spt5 on the clamp[61,62]. These observations explain why Paf1C and Spt4-Spt5 bind the EC only upon disassembly of the initiation complex. These elongation factors appear to bind Pol II weakly in isolation, but interactions between them[32,33] and between Paf1C and TFIIS[45] may enhance their polymerase association. We note that the second contact of Paf1C with Pol II, formed between part C and Rpb3, does not overlap with known factor positions on Pol II. This contact may therefore help to retain Paf1C in the EC when TFIIF reassembles during elongation[63].

Despite our success in obtaining structural information on the highly flexible EC containing Paf1C, the mechanisms Paf1C uses to facilitate chromatin transcription remain unclear. Insights into the mechanisms of chromatin transcription will only come from a structural analysis of the interface of Paf1C with nucleosomes and other chromatin-associated factors. This poses a highly formidable challenge for the future. In the meantime, our results provide the overall Paf1C architecture and location on the polymerase, elucidate the molecular basis for transcription factor exchange during the initiation–elongation transition, and provide a framework for further dissection of the multiple functions of Paf1C in transcription.

## Methods

**Paf1C expression and purification.** The gene sequences encoding for Ctr9, Rtf1 and Cdc73 were amplified from the *S. cerevisiae* genomic DNA by polymerase chain reaction (PCR). Open reading frames of Paf1 and Leo1 were chemically synthesized by GeneArt (Thermo Fisher Scientific) to optimize codon usage for more efficient bacterial expression. Full-length Ctr9 was cloned into pET24b (Novagen) resulting in a non-cleavable hexahistidine tag fused to the Ctr9 carboxyl terminus. The PCR products of full-length Rtf1 or Cdc73 were cloned into an in-house modified version of the pET21b vector (Novagen). Ribosome binding sequence-Cdc73, which contains an ribosome binding sequence on the amino terminus of Cdc73 was inserted sequentially after Rtf1. Additionally, Cdc73 was cloned into multiple cloning sites 2 of pETDuet-1 (Novagen) for the coexpression of a four-subunit Paf1 complex lacking Rtf1 (Paf1C-ΔRtf1). Leo1 and Paf1 were PCR amplified and cloned into two multiple cloning sites of pCDFDuet-1 vector (Novagen) separately and sequentially. The five-subunit Paf1C and its variants were heterologously coexpressed in *E. coli* BL21 CodonPlus(DE3)RIL cells (Stratagene). Cultures of bacteria transformed with pET24b-Ctr9 and pCDFDuet-1-Leo1-Paf1 were grown in Luria–Bertani medium at 37 °C to an $OD_{600 nm}$ of ~0.8. The temperature was reduced to 18 °C, and protein overexpression was induced by addition of 1 mM isopropyl-β-D-thiogalactoside and with continued growth at

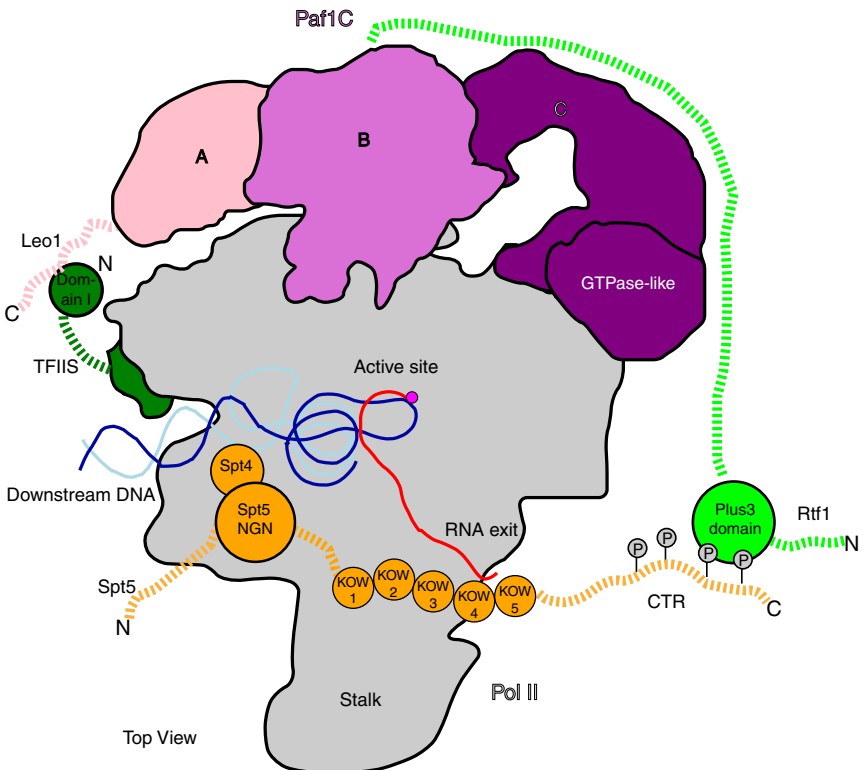

**Figure 7 | Model of the Pol II EC with bound elongation factors.** Summary of our current understanding of elongation factor location and interactions on the Pol II EC surface. In addition to direct Pol II interactions by domains in TFIIS, Spt5 and Paf1C, elongation factors contain many flexible domains and regions that interact. In particular, the Paf1-Leo1 heterodimer may contact the N-terminal domain I of TFIIS and the C-terminal region of Rtf1 is anchored to Paf1C mainly by Paf1 and Ctr9 subunits shown in XL-MS, whereas its Plus3 domain binds the flexible CTR of Spt5. The N-terminal tail of Rtf1 is important for recruiting the chromatin remodeller Chd1 (ref. 9). Dashed lines in this model indicate flexible regions.

18 °C overnight. The same strategy was used for the coexpression of Rtf1 and Cdc73.

Cells were collected and colysed by sonication in buffer A (50 mM Tris (pH 8.7), 600 mM potassium acetate (KOAC), 2 mM dithiothreitol (DTT), 2 mM MgCl₂, 10 mM imidazole, 10 μM ZnCl₂) containing a 1:100 dilution of protease inhibitor cocktail (1 mM leupeptin, 2 mM pepstatin A, 100 mM phenylmethylsulfonyl fluoride, 280 mM benzamidine), 1,000 U benzonase and 0.4 μg ml$^{-1}$ DNaseI. The extract was cleared by centrifugation (20 000$g$, 30 min) and the supernatant was loaded onto a 2-ml Ni-NTA agarose bead column (Qiagen), equilibrated in buffer A. The column was washed extensively with buffer A containing 20 mM imidazole. The complex was eluted with buffer A containing 150 mM imidazole. The eluted protein was diluted 10-fold and was further purified by cation exchange chromatography using a 1-ml HiTrap SP HP column (GE Healthcare). The column was equilibrated in buffer B (50 mM Tris (pH 8.7), 70 mM KOAC, 2 mM MgCl₂, 2 mM DTT, 10 μM ZnCl₂) and proteins were eluted with a linear gradient from 70 mM to 2 M KOAC in buffer B. Fractions containing the protein of interest were concentrated and loaded onto a Superose 6 10/300 (GE Healthcare) size-exclusion chromatography column equilibrated with buffer C (10 mM Tris (pH 8.5), 200 mM NaCl, 2 mM MgCl₂, 2 mM DTT, 10 μM ZnCl₂) or buffer D (10 mM HEPES (pH 7.5), 100 mM NaCl, 2 mM MgCl₂, 2 mM DTT, 10 μM ZnCl₂). The protein complex was concentrated by centrifugation in Amicon Ultra 4-ml concentrators (molecular weight cutoff = 50 kDa; Millipore) to 3 mg ml$^{-1}$. Protein was aliquoted, flash frozen in liquid nitrogen and stored at −80 °C. The similar strategy was used for purifying the five-subunit Paf1C variants.

The four-subunit Paf1 complex (Paf1C-ΔRtf1) and its variants were expressed and purified as described for the five-subunit complex (Paf1C), except that we cotransformed pET24b-Ctr9, pCDFDuet-1-Leo1-Paf1 and pETDuet-1-Cdc73 plasmids together in *E. coli* BL21 CodonPlus (DE3) RIL cells. All the components were separated by SDS–PAGE and confirmed by mass spectrometry.

**Limited proteolysis and Edman sequencing.** Limited proteolysis experiments were performed in buffer C (10 mM Tris (pH 8.5), 200 mM NaCl, 2 mM MgCl₂, 2 mM DTT, 10 μM ZnCl₂) containing 1.6 mg ml$^{-1}$ Paf1C and different concentration of respective protease. A measure of 0.04–0.4 ng μl$^{-1}$ subtilisin, 1–10 ng μl$^{-1}$ trypsin and 2 ng μl$^{-1}$ ArgC were incubated with Paf1C at room temperature for 2 min, 2 min and a 5–20 min time course, respectively. The reactions were stopped using SDS loading buffer. Samples were analysed by SDS–

PAGE. The separated limited proteolysis products on the SDS gel were blotted onto a polyvinylidene difluoride membrane. The membrane was stained with Ponceau S before the fragments were excised and sequenced using a Procise cLC (Applied Biosystems, Foster City, CA, USA).

**Pull-down assay.** Pol II containing a biotin acceptor peptide at the N terminus of Rpb3 was purified. Enzymatic biotinylation was carried out *in vitro* in 100 μl reactions containing 124 μg of Pol II, 8 μg BirA, 2 mM ATP and 100 μM biotin supplemented with dilution buffer (20 mM HEPES (pH 7.5), 16 mM MgCl₂, 10 μM ZnCl₂, 10 mM DTT). The mixture was incubated for 2 h in a thermomixer at 20 °C and 600 r.p.m. Buffer exchange to Pol II buffer (5 mM HEPES (pH 7.5), 10 μM ZnCl₂, 40 mM (NH₄)₂SO₄, 10 mM DTT) was carried out using Micro Bio-SpinTM Chromatography columns (Bio-Rad). In each pull-down assay, 5.8 μg biotinylated Pol II was immobilized on 20 μl Dynabeads M-280 streptavidin resin (Thermo Fisher Scientific), equilibrated in buffer P (50 mM HEPES (pH 7.5), 0.1% NP-40, 150 mM KOAC, 2 mM DTT). Fivefold molar excess of purified Paf1C or/and TFIIS were incubated with immobilized Pol II or control resin at 4 °C for 1 h. Beads were washed five times. Input and the bound proteins were subjected to SDS–PAGE analysis. Relative binding fold changes between Pol II and Paf1 complexes when TFIIS or TFIIS-ΔN130 presents were quantified using Image J. The amount of Leo1 (Paf1C subunit) is quantified relative to the amount of Pol II (subunit Rpb4) in each lane. The fold change is set to 1 in cases lacking TFIIS (representing unchanged amounts). Error bars indicate the s.e.m. from three replicates.

**EC preparation.** The nucleic acid scaffold (Integrated DNA Technologies) used for transcribing mammalian RNA Pol II[48], which contains an 11 nucleotide mismatch transcription bubble and 20 nucleotide RNA (bubble-RNA) was used to assemble Pol II-Paf1C-TFIIS EC (template DNA sequence 5′-AAGCTCAAGTACTTA AGCCTGGTCATTACTAGTACTGCC-3′, non-template DNA sequence 5′-GGCAGTACTAGTAAACTAGTATTGAAAGTACTTGAG CTT-3′ and RNA sequence 5′-UAUAUGCAUAAAGACCAGGC-3′). Equimolar amounts of DNA and RNA were mixed and annealed in a Biometra T3000 Thermocycler that heated to 95 °C and cooled in 1 °C increments every 30 s until 10 °C were reached. The Pol II-bubble-RNA was assembled by incubating 250 pmol purified Pol II with equimolar bubble-RNA in Pol II buffer for 10 min at 25 °C. A 1.8-fold molar excess of Paf1C-Ctr9-ΔC913 and a 1.8-fold molar excess of TFIIS (DE-AA)-inactive mutant were incubated with Pol II-bubble-RNA in assembly buffer D (10 mM

HEPES (pH 7.5), 100 mM NaCl, 2 mM $MgCl_2$, 2 mM DTT, 10 μM $ZnCl_2$) for 15 min at 20 °C in a 65 μl reaction volume. We then centrifuged the reaction for 10 min at 4 °C at 15,000 r.p.m. and kept the supernatant for size-exclusion chromatography or gradient fixation (GraFix) before XL-MS or EM, respectively.

**Crosslinking and mass spectrometry analysis.** The assembled Pol II-Paf1C-TFIIS complex was injected onto size-exclusion chromatography to obtain a homogeneous complex. The fractions containing target complex were collected and crosslinked at various concentrations of BS3 (Thermo Fisher Scientific) to determine empirically the optimal reaction conditions. The best condition, 0.5 mM BS3, was sufficient to convert most of individual component into a high-molecular-weight band in SDS–PAGE and was chosen for final sample preparation. The Pol II-Paf1C-TFIIS at a concentration of 425 μg ml$^{-1}$ was crosslinked with 0.5 mM BS3 and incubated for 30 min at 30 °C. The reaction was quenched by adding 50 mM ammonium bicarbonate. The crosslinked sample was repurified by size-exclusion chromatography on a Superose 6 PC 3.2/300 column (GE Healthcare) equilibrated in buffer D (10 mM HEPES (pH 7.5), 100 mM NaCl, 2 mM $MgCl_2$, 2 mM DTT, 10 μM $ZnCl_2$). Crosslinked sample was digested with trypsin.

Crosslinked peptides were enriched and divided into two halves. Both halves of the sample were measured on an Orbitrap Fusion LC-MS/MS instrumentation platform (Thermo Fisher Scientific) and the data sets were analysed with pLink 1.23 (ref. 64) against a database containing the sequences of the proteins components in the complex separately. An initial false discovery rate cutoff of 1% was set. E-value was calculated in this processing. To visualize this score better, the negative logarithm of E-value was used. The final set of crosslinks were required to satisfy 3 criteria: (1) appeared in both replicates; (2) the max score value from each data set was higher than 5; (3) each crosslink must have a minimal spectral count of 2 in each data set. The final result was subsequently visualized using the xiNET online server[65]. The same strategy was used for Paf1C XL-MS, except that the samples were measured on an AB Sciex Triple-ToF instrument (AB SCIEX).

**Gradient fixation.** To reconstitute a homogeneous Pol II-Paf1C-TFIIS EC, the sucrose GraFix was carried out as described[50]. Each sucrose gradient for GraFix was generated by mixing equal volumes of light solution (10 mM HEPES (pH 7.5), 100 mM NaCl, 2 mM DTT, 2 mM $MgCl_2$, 10 μM $ZnCl_2$, 10% (w/v) sucrose) and heavy solution 1 (10 mM HEPES (pH 7.5), 100 mM NaCl, 2 mM DTT, 2 mM $MgCl_2$, 10 μM $ZnCl_2$, 30% (w/v) sucrose, 0.075% (w/v) glutaraldehyde) using a gradient mixer (Gradient Master 108; BioComp Instruments). This resulted in a dual gradient of 10–30% sucrose and 0–0.075% glutaraldehyde in an 11 × 60 mm$^2$ ultracentrifuge tube (Beckman Coulter). Next, 60 μl of the in vitro reconstituted Pol II-Paf1C-TFIIS EC were applied on top of the gradient. After ultracentrifugation at 32,000 r.p.m. in a SW60 swinging bucket rotor (Beckman Coulter) for 16 h at 4 °C, 200 μl fractions of the gradient were collected by pipetting carefully from top to bottom of the tube. Parallel sucrose gradient fractions of samples applied to gradients either containing or lacking glutaraldehyde showed the same sedimentation profile when analysed by SDS–PAGE. The crosslinking reaction was quenched by adding 0.5 M (pH 7.8) aspartate to a final concentration of 12.5 mM. Micro spin chromatography columns (Bio-Rad) were used for buffer exchange to remove sucrose, glutaraldehyde and aspartate. The samples were concentrated using a GE concentrator (molecular weight cutoff = 100 kDa; GE Healthcare) and immediately used for cryo-EM.

**Negative stain EM.** Pol II-Paf1C-TFIIS EC was prepared as described above. Negative staining of the specimen was carried out as described[48]. Micrographs were obtained using an FEI Philips CM-200 at 160 kV with a original magnification of × 88 000 (corresponding to a calibrated sampling of 2.51 Å per physical pixel) and a defocus ∼ 2.0 μm.

**Cryo-EM specimen preparation and data acquisition.** An FEI Vitrobot Mark IV plunger (FEI) was used for preparation of frozen-hydrated specimens. Four microlitres of sample was placed onto Quantifoil Cu R3.5/1 and Cu R2/1 glow-discharged 200 mesh holey carbon grids, which were then blotted for 8.5 s with blot force 13 to remove the excess solution before they were flash frozen in liquid ethane. The Vitrobot chamber was operated at constant 4 °C and 100% humidity during blotting. The grids were transferred and stored in liquid nitrogen before data acquisition.

Two cryo-EM data sets were acquired on a 300 keV FEI Titan Krios electron microscope equipped with a K2 Summit direct electron counting camera (Gatan) positioned post a GIF Quantum energy filter (Gatan) to increase the signal-to-noise ratio. Automated data collection was carried out using the TOM toolbox[66]. Movie images were recorded at a nominal magnification of × 37,000 (corresponding to a calibrated sampling of 1.35 Å per physical pixel) in super-resolution mode, thus yielding a pixel size of 0.675 Å per pixel. For the first data set, two movies were acquired in each hole and a total of 595 movie stacks with a defocus range of − 0.7 to − 4.2 μm were collected from Quantifoil Cu R3.5/1 grids at a dose rate of 7.6 electrons per pixel per second. Each movie encompassed a total dose of ∼33 electrons per Å$^2$ with a total exposure time of 10.8 s fractionated into 27 frames. Each frame had an exposure time of 0.4 s. For the second data set collected from the Quantifoil Cu R2/1 grids, one movie was acquired in each hole and 2,146 movie

stacks were recorded with an exposure time of 12 s fractionated into 30 frames, a dose rate of 4.2 electrons per pixel per second. Each frame had an exposure time of 0.4 s, resulting in a total accumulated dose of ∼ 28 electrons per Å$^2$ per stack. Defocus values ranged from − 0.6 to − 4.2 μm. Movie stacks from two data sets were aligned and binned as previously described with the frame-based motion-correction algorithm to generate drift-corrected micrographs for further processing[67], except images were not partitioned into quadrants.

**Cryo-EM image processing.** Contrast transfer function was estimated using CTFFIND3 (ref. 68) and CTFFIND4 (ref. 69) for the data sets from Quantifoil Cu R3.5/1 grids (R3.5) and Quantifoil Cu R2/1 grids (R2), respectively. For the R3.5 data set, 123 aligned micrographs were excluded because of contaminations or bad ice quality. After removing these micrographs, we used e2boxer.py from EMAN2 package[70] to semiautomatically pick 84,362 particles with a box size of 240$^2$ pixels from the remaining micrographs. 2D reference-free classification within RELION 1.4 (ref. 51) was used to remove micelles or other false positives. After this step 79,024 particles remained. We deleted 512 aligned micrographs from the R2 data set and semiautomatically picked 21,301 particles using e2boxer.py from EMAN2 package with a box size of 240$^2$ pixels.

Reference-free 2D classes were generated, and nine representative classes were low-pass filtered to 20 Å and used as templates for autopicking[71]. The resulting 863,235 particles were screened manually and subjected to reference-free 2D classification, yielding 291,817 particles. We then merged particles from two datasets. 370,841 particles were used as an input for the subsequent 3D reconstruction performed with RELION 1.4 (ref. 51), unless noted otherwise. A published reconstruction of bovine Pol II[48] filtered to 50 Å was used as an initial reference for 3D refinement. The aligned particles were subjected to particle polishing using RELION 1.4 (ref. 72) to reduce the noise and correct the local motion and radiation damage.

3D classifications were carried out without image alignment. The first 3D classification was performed to separate out particles lacking the Paf1C density. This led to the dismissal of 158,422 particles. We refined the remaining 212,419 Paf1C-containing particles. Gold-standard FSC was calculated after the 3D refinement in RELION. A 5.5 Å average resolution map was observed at the gold-standard FSC criteria of 0.143. In the second step, a soft mask encompassing Paf1C was generated using the 'volume erase' option in UCSF Chimera[53] and RELION[51]. After 3D classification within this mask, we obtained three Pol II-Paf1C-TFIIS maps resolving different parts of Paf1C density (parts A, B and C). The individual classes were autorefined using the respective masks. The complex containing part A was refined from 114,672 particles to an average resolution of 5.7 Å. The complex containing part B was refined from 54,722 particles to an average resolution of 5.9 Å. The complex containing part C was refined from 43,025 particles to an average resolution of 6.2 Å. All three maps were unsharpened and filtered according to local resolution[73]. All structural figures were generated using UCSF Chimera[53] and PyMOL (Schrödinger LLC.).

**Structural modelling.** The X-ray crystallographic structure of the Pol II-TFIIS complex (PDB entry: 3PO3; ref. 49) was used as the starting reference model and placed into the cryo-EM map using UCSF Chimera[53]. Amino acids 68–89 and 132–168 of the protrusion domain of Rpb2 were replaced by the coordinates from the PDB entry 5C4J[52]. Template DNA (nucleotides 1–33), non-template DNA (nucleotides 7–39) and RNA (nucleotides 7–20) were derived from the PDB entry 5FLM (ref. 48). The models were fitted as rigid bodies into the density map using 'Fit in Map' in UCSF Chimera[53]. The C-terminal GTPase-like domain of Cdc73 (PDB entry: 3V46) was fitted into the map of the complex containing part C using UCSF Chimera[53] and adjusted in COOT[74] based on the XL-MS results. We also calculated the approximate molecular weight of the density, which is present in addition to Pol II, with Chimera using TFIIS as a reference[53].

**Analytical sucrose gradient ultracentrifugation.** To investigate the binding of TFIIF and Paf1C to Pol II, analytical sucrose gradient ultracentrifugation was carried out. A sucrose gradient was generated by mixing equal volumes of light solution (10 mM HEPES (pH 7.5), 100 mM NaCl, 2 mM DTT, 2 mM $MgCl_2$, 10 μM $ZnCl_2$, 10% (w/v) sucrose) and heavy solution 2 (10 mM HEPES (pH 7.5), 100 mM NaCl, 2 mM DTT, 2 mM $MgCl_2$, 10 μM $ZnCl_2$, 30% (w/v) sucrose) using a gradient mixer (Gradient Master 108; BioComp Instruments). Reconstituted Pol II-Paf1C-TFIIS EC was divided into portions. The first portion was incubated with a 1.8-fold molar excess of TFIIF[54], while the other portion was incubated with buffer D (10 mM HEPES (pH 7.5), 100 mM NaCl, 2 mM DTT, 2 mM $MgCl_2$, 10 μM $ZnCl_2$) as a control. Next, the reconstituted complexes were applied on top of the gradient. After ultracentrifugation at 32,000 r.p.m. in a SW60 swinging bucket rotor (Beckman Coulter) for 16 h at 4 °C, 200 μl fractions were collected by pipetting carefully from top to bottom of the tube before analysing them by SDS–PAGE and native PAGE. The assay was repeated but starting from a Pol II-TFIIF complex and then incubating with excess Paf1C-TFIIS.

**4tU-Seq data collection.** 4tU labelling of cellular RNA was performed as described[75] with minor changes. Forty microlitres of each replicate culture were

used for metabolic RNA labelling. The strains used in this study paf1Δ (MATa his3Δ1 leu2Δ0 met15Δ0 ura3Δ0 paf1::kanMX) and rtf1Δ (MATa his3Δ1 leu2Δ0 met15Δ0 ura3Δ0 rtf1::kanMX) were purchased from Euroscarf. Yeast cells were grown in yeast extract peptone dextrose medium overnight, diluted to an $OD_{600\,nm}$ of 0.1 and grown to an $OD_{600\,nm}$ of 0.8. 4tU (Sigma; 2 M in dimethylsulfoxide) was added to the media at a final concentration of 5 mM, and cells were collected after 6 min of labelling by centrifugation at 2,465g and 30 °C for 1 min. The supernatant was discarded and the pellet resuspended in RNAlater solution (Ambion/Applied Biosystems) and then flash frozen in liquid nitrogen. Cell number was determined from an aliquot with a Cellometer N10 (Nexus) cell counter. Total RNA was extracted with phenol chloroform. RNA spike-ins were added to cell pellets at the first step of RNA purification[58]. Briefly, six artificial RNAs (spike-ins: ERCC-00043, ERCC-00170, ERCC-00136, ERCC-00145, ERCC-00092 and ERCC-00002) from the ERCC RNA spike-in mix (Life Technologies) were mixed in equal amount to all samples. Each spike-in was subjected to in vitro transcription beforehand with the Megascript T7 Transcription Kit (Life Technologies) with either 1:10 4sUTP:UTP ratio for spike-ins ERCC-00043, ERCC-00136 and ERCC-00092 or only UTP for spike-ins ERCC-00170, ERCC-00145 and ERCC-00002; resulting in labelled and non-labelled spike-ins, respectively. The amount of spike-ins was adjusted to the cell number of each sample (120 ng of spike-in mix to $2.5 \times 10^8$ cells for all samples). Labelled RNA was chemically biotinylated and purified using strepatavidin-coated magnetic beads as described[76]. Sequencing libraries were prepared according to the manufacturer's recommendations using the Ovation Universal RNA-Seq System (NuGEN). Libraries were quantified with Qubit 1.0. Libraries were pooled and sequenced on an Illumina HiSeq 2500 sequencer.

**4tU-Seq data analysis.** Data analysis was performed as described[57], with minor modifications. Briefly, paired-end 50 bp reads with additional 6 bp of barcodes were obtained for labelled RNA. Reads were demultiplexed and aligned to the S. cerevisiae genome (sacCer3, version 64.2.1) using STAR (version 2.3.0)[77]. SAMTools was used to quality filter SAM files[77]. Alignments with MAPQ smaller than 7 ($-q$ 7) were skipped and only proper pairs ($-f99$, $-f147$, $-f83$, $-f163$) were selected. Further processing of the RNA-Seq data was carried out using the R/Bioconductor environment. Piled-up counts for every genomic position were summed up over replicates, using physical coverage, that is, counting both sequenced bases covered by reads and unsequenced bases spanned between proper mate-pair reads. We used a spike-in (RNAs) normalization strategy as described before[58] to allow for observation of global changes in the 4sU-Seq signal. Read counts for all spike-ins necessary to that end were calculated using HTSeq[78] after mapping. To estimate these normalization factors, we used a statistical model that describes the read counts for each gene in each sample by gene-specific amounts of labelled RNA. The model includes a parameter for the length of the respective feature. The expectation of the number of reads is fitted by maximum-likelihood assuming negative binomial distribution with dispersion parameters as calculated by DESeq[79].

**Data availability.** The cryo-EM data has been deposited in Electron Microscopy Data Bank (EMDB) under accession codes EMD-3626, EMD-3627, EMD-3628 and EMD-3629 for the RNA polymerase II-Paf1C-TFIIS, RNA polymerase II-Paf1C-TFIIS-A, -B and -C structures, respectively. The sequencing data and the annotation file have been deposited in the Gene Expression Omnibus database under accession code GSE95556. The data that support the findings of this study are available from the corresponding author on request.

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

## Acknowledgements

We thank Thomas Schulz for yeast fermentation, Carina Burzinski for Pol II and TFIIF purification, Thomas Fröhlich (LAFUGA, Gene Center Munich) and Monika Raabe from the Urlaub laboratory for protein identification, Romina Hofele from the Urlaub Laboratory for Paf1C XL-MS analysis, Simon Neyer for help with EM data processing and suggestions on the manuscript. P.C. was supported by the Deutsche Forschungsgemeinschaft (SFB860, SPP1935), the Advanced Grant 'TRANSREGULON' from the European Research Council (Grant Agreement No. 693023) and the Volkswagen Foundation.

## Author contributions

Y.X. designed and performed experiments, unless stated otherwise. C.B. helped with EM data collection. J.M.P. provided access to the EM facility and supervised EM data collection. C.-T.L. performed MS analysis. H.U. supervised MS analysis. K.C.M. carried out the 4tU-Seq experiment. B.S. analysed 4tU-Seq data. C.B. and D.T. helped with EM data processing. P.C. designed and supervised research. Y.X. and P.C. prepared the manuscript, with the input from all authors.

## Additional information

**Competing interests:** The authors declare no competing financial interests.

