## [Peer Review File · Nature Communications]

Reviewers' Comments:

Reviewer #1 (Remarks to the Author):

This manuscript presents structural and biochemical studies of the Paf1C multi-protein complex (Paf1, Leo1, Ctr9, Rtf1, Cdc73) from *S. cerevisiae*: a Pol II transcription elongation factor with additional roles in chromatin remodeling and mRNA processing. First, the recombinant bacterial expression and purification of the Paf1C complex is shown, and pull-down experiments demonstrate an interaction with Pol II, which is enhanced in the presence of TFIIS. Next, the cryo-EM structure of a crosslinked Pol II-Paf1C-TFIIS complex is presented. The density for Paf1C is a low-resolution composite of three independently-refined 3D classes, and to aid interpretation, chemical crosslinking coupled to mass spectrometry (XL-MS) is performed. Because the observed density for Paf1C clashes with the known structure of the Pol II-TFIIF initiation complex, the authors propose a model in which TFIIF and Paf1C compete for binding to Pol II, and conduct assays to demonstrate this. Finally the authors demonstrate the importance of Paf1C in vivo by using 4tU-Seq experiments to show a global decrease in mRNA synthesis in Δ Paf1 and Δ Rtf1 yeast strains.

In general, the paper is well written and figures are clearly presented, and the topic is of particular interest to the transcription and RNA processing communities. This complex was obviously technically very challenging and did not lend itself well to structure determination by cryo-EM; such an ambitious effort is therefore commendable. Overall, the main finding is identification of Paf1C binding sites on Pol II, and how Paf1C binding impacts other Pol II interaction partners. However, too much emphasis is placed on the cryoEM structure and several key conclusions are not justified based on the presented data. The results and discussion should be re-written in light of this, and further experiments should be performed to validate and extend the structural aspects of the work. In particular, the following points should be addressed.

Major:

1. The assertion that the Paf1C – Pol II interaction is “strongly enhanced by TFIIIS” (line 134) is not supported by the data. Based on Fig. S2a, comparing lanes 4 and 5 this appears to be the case, but in Fig. S2b, the equivalent lanes 6 and 8 show no such difference. Similarly, in Fig. S2c (lanes 6-9) there isn’t a striking difference in binding when TFIIIS is present (line 132). This leads me to question the reproducibility of these pull-downs. Technical replicates accompanied by band quantification and an analysis of statistical significance would clarify this.

Related to the point above, the authors state that the TFIIIS- Δ N130 construct “could not enhance the Pol II-Paf1C interaction” (line 128). However, in Fig. S2b lane 6 and 7 appear to show that TFIIIS- Δ N130 actually inhibits the interaction slightly. Again, it would be good to see replicates to know if this small difference is statistically significant.

2. The “three-lobed” (A, B, C) model presented in Fig. 3, 5 and 7 is not actually a cryo-EM structure. It is a composite of three structures of Pol II, each with a different “lobe”, obtained by 3D classification - the real structural result of the paper is shown in Supplementary Figures 4 and 5. It is therefore potentially misleading to combine the three “lobes” and present them as continuous Paf1C density without any direct structural evidence that this is the case. (e.g. why do lobes A and B and C never occur in the same 3D class?). The authors state that “their relative orientation is flexible...due to mobility on the Pol II surface” (line 170) – if this is true, the relevance of the presented structure is unclear. During the discussion of XL-MS data, the authors state (line 234) “Ctr9 did not crosslink effectively to Pol II, consistent with the bridging density B that does not reach the Pol II surface.” If density B does not contact Pol II, how can it form a stable 3D class in the absence of densities A and C providing the Pol II interactions?

One possible explanation for A, B and C densities being observed independently is that the complex falls apart during vitrification of EM grids – a common problem in the field. I assume that stability was a problem because crosslinking was performed prior to cryo-EM. A negative stain EM dataset should provide a low-resolution 3D reconstruction that would help validate the cryo-EM maps and demonstrate that lobes A, B and C actually do form continuous density corresponding to Paf1C. Indeed, the authors state that negative stain EM was performed (line 144) but never show any micrographs or 2D classes. A negative stain 3D model would make the proposed cryo-EM model much more convincing, given the problems mentioned above.

3. Whilst the XL-MS gives valuable insight into the Paf1C-Pol II interconnectivity, there is insufficient evidence for assigning any proteins to the cryo-EM densities without more evidence

that the densities can co-exist in a continuous manner, especially since the masses of the individual proteins do not correspond to the sizes of the observed lobes. As such, the discussion in lines 235 – 242 should be removed or substantially reworded.

4. The authors conclude from sucrose gradient experiments (Fig. 5) that Paf1C and TFIIF bind to Pol II in a competitive manner. However, my interpretation of Fig. 5 is that TFIIF binds more tightly to Pol II and can displace Paf1C, whereas Paf1C cannot effectively displace TFIIF. This explanation is actually more consistent with the idea that Paf1C lobe A region is flexible and can occupy multiple positions on the Pol II surface – one would not expect such a protein to be able to compete off a tightly bound factor (TFIIF) at a specific site. Pull-down experiments with different competitor concentrations could help clarify the relative affinities and show whether binding is really mutually exclusive.

5. The 4tU-seq experiments presented in Fig. 6 effectively show that Δ Paf1 and Δ Rtf1 strains have defects in mRNA synthesis. However, the authors do not show that this decrease is unique to Pol II transcripts. Control analyses looking at relative abundance of Pol III transcripts in the WT and knockout strains would strengthen this.

Minor:

1. The statement is often made that the five Paf1C subunits are present in approximately stoichiometric amounts, yet Leo1 often appears sub-stoichiometric (e.g. Fig. 2). What is the reason for this? It is surprising as this is the subunit that crosslinks to Rpb2 the most. Fig 4a shows that most of the Pol II-Paf1C crosslinks map to a 100 amino acid stretch of Leo1. This is the same region of Leo1 that forms most of the intra-Paf1C crosslinks also. Is this region of Leo1 particularly notable, e.g. disordered/flexible/lysine-rich? This relative crosslinking promiscuity (compared to the other Paf1C subunits) should be discussed more in the text.

2. It was unclear what the rationale was for making the various truncation mutations, and how the mutations were chosen (Lines 115 – 118). Were these constructs necessary to optimize expression/improve solubility/reduce degradation? The pull-down experiments in Supplementary Fig. 2 were not conducted with the core Paf1C.

A reference to Fig S1 on Line 176 would help.

3. Supplementary Fig. 3 seems like a logical extension of Fig. 2, and could be combined into the same figure. The legend in this figure should include more details about the conditions of image acquisition e.g. K2 camera, magnification, dose.

It would also be good to include representative negative stain micrographs, 2D classes and a 3D reconstruction here, or as a separate figure.

4. The individual lobe structures presented in Supplementary Fig. 4 or 5 should be moved to the main article, in Fig. 3, as these are the main EM results of the paper. The merged “three-lobed” structures shown in Fig. 3, 5 and 7 represent an interpretation of the three separate structures and should be defined as such. Do volumes A, B and C clash or overlap with each other when merged to generate the composite?

5. To present the cross-linking data (Figure 4d), it would be better to show the crystal structure of Pol II with crosslinked residues highlighted. This would allow the reader to identify surface patches that interact with each of the Paf1C subunits. The locations of the three separate cryo-EM lobes could be indicated beside the structure as panels allowing visualization of the relative proximity of the crosslinked Pol II residues to the separate blobs of EM density.

6. The phrase “movie stacks” should be replaced with “micrographs” (line 145).

7. The colors in Fig. 5a should be changed: Both Paf1C density and TFIIIF subunits are different shades of pink/purple, which is confusing.

8. The resolutions listed in lines 156 and 160 are misleading, as this only refers to the Pol II core (see Supplementary Fig. 5e-g). The actual resolutions for densities A, B and C are more like 10 – 20 Å. This should be clearly stated in the text.

9. Referring to Cdc73, the authors state “we could fit this crystal structure to the globular density” (line 227), but the actual fit is not shown (apart from an outline in Fig. 4d.) A figure should be included to show this fit in 3D.
10. The statement in the discussion “thus our results elucidate the general transcription cycle” (line 289) should be removed.
11. Typo (line 724) “starting form”
12. In the legend for Fig. 3 (line 844) the authors state “The same threshold level in Chimera was used to contour densities”. It is unclear to what densities this refers – A, B, C and Pol II? Or just A, B and C? It looks like Pol II is displayed at a different threshold to the other densities.
13. In Fig. 4d, it would be useful to include an approximate measure of length on the indicated crosslinks.

Reviewer #2 (Remarks to the Author):

In this study, Xu et al. use cryo-electron microscopy, chemical crosslinking/mass spectrometry, and protein binding studies to investigate the structure of the yeast Paf1C bound to Pol II and TFIIIS. Paf1C has been implicated in regulating chromatin structure during transcription elongation but little is known about this conserved complex at the structural level. Through their cryo-EM analysis, the authors provide the first structural insights into the Paf1 complex and its interactions with Pol II. This is an important contribution to the transcription/chromatin field, as previous crystallography studies focused on small functional domains in Paf1C. Other conclusions from this work indicate that the elongation factor TFIIIS can enhance Paf1C association with Pol II and that the initiation factor TFIIIF may preclude binding of Paf1C to Pol II through a steric clash. Nascent transcript profiling studies support an important role for Paf1C in global regulation of transcription; however, the impact of this part of the paper is limited by the analysis that was performed. Overall, the primary strength of this paper is the cryo-EM structure of Paf1C and its association with Pol II and TFIIIS. The other contributions of the paper

are weaker primarily because the data indicate somewhat subtle effects that were analyzed by only a single approach or because the results were not fully analyzed.

1. The introduction is not very well written. It reads like a list of facts with insufficient cohesion.
2. The supplemental figures are cited out of order. The figures should be re-formatted accordingly.
3. There is insufficient information on how the limited proteolysis data were analyzed to define the core Paf1C.
4. The term “synergistic” seems inappropriate when describing enhanced binding of Paf1C in the presence of TFIIIS. Does Paf1C enhance the binding of TFIIIS?
5. Supplementary Figure 2C: The authors report enhanced binding of Paf1C derivatives in the presence of TFIIIS. This is very difficult to see in the figures provided. The bands in this and other gels are faint, making comparisons very difficult. Putting the overall faintness of the gel aside, I do not see a significant difference in Paf1C subunit levels between reactions containing TFIIIS and those lacking TFIIIS. Ctr9 and Cdc73 levels look nearly identical among the four samples in lanes 6-9. Moreover, coomassie staining is not a very quantitative measure of protein abundance.
6. The assignment of specific subunits to the three Paf1C parts (A, B, and C) was guided by chemical crosslinking data. Do the authors have any other information to verify the assignments? For example, Ctr9 is assigned to part B because it crosslinks to Paf1/Leo1 (assigned to A) and Cdc73 (assigned to B). This seems logical but the same argument could be made for Rtf1, which is not assigned to any part of the structure.
7. Can the authors fit the Paf1-Leo1 crystal structure into Paf1C part A?
8. Qiu et al. reported that Cdc73 binds to the CTD of Rpb1. The authors do not address this interaction in their structure or in their description of the model.
9. The conclusion that binding of TFIIIF to Pol II impedes the binding of Paf1C due to a steric clash is intriguing and supports the notion that transition from initiation to elongation is controlled by an exchange of Pol II-bound factors. However, the data for this are not especially strong. In panels 5B and 5C, binding of Paf1C and TFIIIF do not appear mutually exclusive. Moreover, the staining is faint and not quantitative. In the absence of additional biochemical or functional data, the conclusion of a competition between TFIIIF and Paf1C is not yet convincing. A more minor point: the figures could be made clearer by using colors other than shades of purple/pink for both Paf1C and TFIIIF.

10. The 4tU-seq data do not add very much to the paper. They are under-analyzed and appear to be tacked on. I would recommend omitting the data or providing a more complete analysis. Also, do the authors have an explanation for why the *rtf1* mutant would show a greater decrease in mRNA levels than the *paf1* mutant? For many phenotypes, *paf1* mutants are more severely compromised than *rtf1* mutants.
11. How was the spike-in normalization carried out? More details would be helpful.
12. Line 58: “Paf1C” should replace “Paf1”
13. Lines 203-206: The finding of a crosslink between Leo1 and TFIIS is new and interesting. However, with the current writing, the authors suggest it provides a “positive control”.
14. Line 214: What is the evidence for a “subassembly” involving Paf1-Leo1-Ctr9?
15. Figure 1: The black underlines are difficult to see. The diagram of Cdc73 should be corrected to read “GTPase-like domain”.
16. Figure 2A: What do the curved lines at the bottom of the marker lane represent?
17. Supplementary Figure 1a legend: “Crt9” should be changed to “Ctr9”.
18. Abstract: Although Paf1C is clearly important for mRNA transcription, the term “required for mRNA transcription” is a bit strong. Yeast cells lacking the complex are viable.

Detailed list of responses to reviewer concerns

NCOMMS-16-27989-T

Xu et al. "Architecture of the RNA polymerase II-Paf1C-TFIIS transcription elongation complex"

Responses are in italics

Editorial concerns:

We have made the best possible effort to improve the discussion of the biological insights gained as recommended by the referees. We have also ensured the formatting is correct. Further, we include a copy of the reporting checklist.

Reviewer #1 (Remarks to the Author):

This manuscript presents structural and biochemical studies of the Paf1C multi-protein complex (Paf1, Leo1, Ctr9, Rtf1, Cdc73) from *S. cerevisiae*: a Pol II transcription elongation factor with additional roles in chromatin remodeling and mRNA processing. First, the recombinant bacterial expression and purification of the Paf1C complex is shown, and pull-down experiments demonstrate an interaction with Pol II, which is enhanced in the presence of TFIIS. Next, the cryo-EM structure of a crosslinked Pol II-Paf1C-TFIIS complex is presented. The density for Paf1C is a low-resolution composite of three independently-refined 3D classes, and to aid interpretation, chemical crosslinking coupled to mass spectrometry (XL-MS) is performed. Because the observed density for Paf1C clashes with the known structure of the Pol II-TFIIF initiation complex, the authors propose a model in which TFIIF and Paf1C compete for binding to Pol II, and conduct assays to demonstrate this. Finally the authors demonstrate the importance of Paf1C in vivo by using 4tU-Seq experiments to show a global decrease in mRNA synthesis in Δ Paf1 and Δ Rtf1 yeast strains.

In general, the paper is well written and figures are clearly presented, and the topic is of particular interest to the transcription and RNA processing communities. This complex was obviously technically very challenging and did not lend itself well to structure determination by cryo-EM; such an ambitious effort is therefore commendable. Overall, the main finding is identification of Paf1C binding sites on Pol II, and how Paf1C binding impacts other Pol II interaction partners. However, too much emphasis is placed on the cryoEM structure and several key conclusions are not justified based on the presented data. The results and discussion should be re-written in light of this, and further experiments should be performed to validate and extend the structural aspects of the work. In particular, the following points should be addressed.

We thank the reviewer for the kind words and helpful comments.

Major:

1. The assertion that the Paf1C – Pol II interaction is “strongly enhanced by TFIIS” (line 134) is not supported by the data. Based on Fig. S2a, comparing lanes 4 and 5 this appears to be the case, but in Fig. S2b, the equivalent lanes 6 and 8 show no such difference. Similarly, in Fig. S2c (lanes 6-9) there isn’t a striking difference in

binding when TFIIS is present (line 132). This leads me to question the reproducibility of these pull-downs. Technical replicates accompanied by band quantification and an analysis of statistical significance would clarify this.

Related to the point above, the authors state that the TFIIS- Δ N130 construct “could not enhance the Pol II-Paf1C interaction” (line 128). However, in Fig. S2b lane 6 and 7 appear to show that TFIIS- Δ N130 actually inhibits the interaction slightly. Again, it would be good to see replicates to know if this small difference is statistically significant.

We observed highly similar results when we repeated these pull-down assays. To make it clear, we quantified the bands using Image J to visualize the change. We now added quantified Leo1/Rpb4 fold change in Supplementary Fig. 2. In Fig. S2d, error bars indicate the standard error of the mean from 3 replicates.

2. The “three-lobed” (A, B, C) model presented in Fig. 3, 5 and 7 is not actually a cryo-EM structure. It is a composite of three structures of Pol II, each with a different “lobe”, obtained by 3D classification - the real structural result of the paper is shown in Supplementary Figures 4 and 5. It is therefore potentially misleading to combine the three “lobes” and present them as continuous Paf1C density without any direct structural evidence that this is the case. (e.g. why do lobes A and B and C never occur in the same 3D class?). The authors state that “their relative orientation is flexible...due to mobility on the Pol II surface” (line 170) – if this is true, the relevance of the presented structure is unclear. During the discussion of XL-MS data, the authors state (line 234) “Ctr9 did not crosslink effectively to Pol II, consistent with the bridging density B that does not reach the Pol II surface.” If density B does not contact Pol II, how can it form a stable 3D class in the absence of densities A and C providing the Pol II interactions?

One possible explanation for A, B and C densities being observed independently is that the complex falls apart during vitrification of EM grids – a common problem in the field. I assume that stability was a problem because crosslinking was performed prior to cryo-EM. A negative stain EM dataset should provide a low-resolution 3D reconstruction that would help validate the cryo-EM maps and demonstrate that lobes A, B and C actually do form continuous density corresponding to Paf1C. Indeed, the authors state that negative stain EM was performed (line 144) but never show any micrographs or 2D classes. A negative stain 3D model would make the proposed cryo-EM model much more convincing, given the problems mentioned above.

We apologize that this may have led to misunderstandings. In our initial 3D reconstruction of the Pol II-Paf1C-TFIIS complex from 57% of the cleaned data set, we observe, at lower resolution, continuous density for all 3 parts of the Paf1C. To clarify this, we have added the continuous density to Fig. 2. In Fig. 2C, a global reconstitution of Pol II-Paf1C-TFIIS EC filtered to 18 Å resolution was shown, which can explain that density A-C is present in the same map. To get better density of each lobe, we performed sub-3D classification, and later combined the densities into a composite map. Thus lobes A, B, and C are not mutually exclusive, but rather we are enriching for particles with increased density (decreased mobility) in a particular region, likely due to lower mobility of these regions in subclasses of particles. Indeed, we see that part A and B have overlapping regions, and part C overlaps with A and B.

We have further changed the notion “Ctr9 did not crosslink effectively to Pol II, consistent with the bridging density B that does not reach the Pol II surface” to “Ctr9 did not crosslink effectively to Pol II, consistent with the bridging density B that forms only limited contacts with the Pol II surface.” This will account for the minor concern the reviewer had.

We also now have added a micrograph from the negative stain analysis to supplementary Fig. 3. On this micrograph we see nice single particles, which shows the high quality of our sample. We then subjected this sample to cryo-EM. We now show our low-resolution model from Cryo-data, which shows the continuous Paf1C density. Compare above for details.

3. Whilst the XL-MS gives valuable insight into the Paf1C-Pol II interconnectivity, there is insufficient evidence for assigning any proteins to the cryo-EM densities without more evidence that the densities can co-exist in a continuous manner, especially since the masses of the individual proteins do not correspond to the sizes of the observed lobes. As such, the discussion in lines 235 – 242 should be removed or substantially reworded.

In the global reconstruction shown in Fig. 2C (see above) there is a continuous density of Paf1C, indicating that the densities coexist. Indeed, it is difficult to assign any individual proteins to the density lobes due to the flexibility of the subunits and the low resolution, but the XL-MS does indicate the location of Paf1C regions within these densities.

4. The authors conclude from sucrose gradient experiments (Fig. 5) that Paf1C and TFIIF bind to Pol II in a competitive manner. However, my interpretation of Fig. 5 is that TFIIF binds more tightly to Pol II and can displace Paf1C, whereas Paf1C cannot effectively displace TFIIF. This explanation is actually more consistent with the idea that Paf1C lobe A region is flexible and can occupy multiple positions on the Pol II surface – one would not expect such a protein to be able to compete off a tightly bound factor (TFIIF) at a specific site. Pull-down experiments with different competitor concentrations could help clarify the relative affinities and show whether binding is really mutually exclusive.

We agree with the reviewer that the binding of TFIIF to Pol II is stronger than the binding of Paf1C to Pol II. To directly show this, we have now included a native PAGE gel to analyze sucrose gradient fractions. This shows that TFIIF can almost entirely compete out the Pol II-bound Paf1C, but not the other way around. We thank the reviewer for the suggestion and think this aspect is now much improved and clarified. We nevertheless agree with the reviewer that the binding may not be mutually exclusive, in particular when other factors are additionally present in cells, and we have modified the text accordingly.

5. The 4tU-seq experiments presented in Fig. 6 effectively show that Δ Paf1 and Δ Rtf1 strains have defects in mRNA synthesis. However, the authors do not show that this decrease is unique to Pol II transcripts. Control analyses looking at relative abundance of Pol III transcripts in the WT and knockout strains would strengthen this.

We analyzed the tRNA transcripts made by Pol III. The tRNA transcripts show a decreased in Δ Paf1 or Δ Rtf1 stains comparing to wild-type. The result is shown below.

However, we cannot distinguish direct from indirect transcriptional effects on Pol III transcripts in the knock out strains. Therefore we refrained to discuss this further in the manuscript.

Minor:

1. The statement is often made that the five Paf1C subunits are present in approximately stoichiometric amounts, yet Leo1 often appears sub-stoichiometric (e.g. Fig. 2). What is the reason for this? It is surprising as this is the subunit that crosslinks to Rpb2 the most. Fig 4a shows that most of the Pol II-Paf1C crosslinks map to a 100 amino acid stretch of Leo1. This is the same region of Leo1 that forms most of the intra-Paf1C crosslinks also. Is this region of Leo1 particularly notable, e.g. disordered/flexible/lysine-rich? This relative crosslinking promiscuity (compared to the other Paf1C subunits) should be discussed more in the text.

Full-length Leo1 is not as stable as the other proteins in the Paf1C. During sample preparation, Leo1 degrades, especially after the gradient centrifugation. There are Leo1 degradation bands visible between bands for Paf1 and Cdc73, which have been confirmed by mass spectrometry. From the protein sequence, the 100-amino acid stretch of Leo1, which has many crosslinks to Pol II and Paf1C, is a lysine-rich region. We added a note on this to the text.

2. It was unclear what the rationale was for making the various truncation mutations, and how the mutations were chosen (Lines 115 – 118). Were these constructs necessary to optimize expression/improve solubility/reduce degradation? The pull-down experiments in Supplementary Fig. 2 were not conducted with the core Paf1C.

The original idea for the various truncations was to remove flexible regions for crystallization trials. First we purified the full-length Paf1C. Then the limited proteolysis coupled to Edman sequencing showed the C-terminal region of Ctr9 is not stable. Then we prepared the Paf1C-Ctr9- Δ 913 complex. Comparing to the full-length complex, it expresses better and shows less degradation. The limited proteolysis also showed the unstable Leo1 and Paf1, which we used in our iterative truncation strategy, and thus prepared the core Paf1C.

When we decided to solve the structure using Cryo-EM, we chose the Paf1C-Ctr9- Δ 913 because of the increased yield and stability compared with the full-length Paf1C,

as mentioned in the manuscript. We observed a similar Pol II-Paf1C interaction with this variant. Taken together, we have used the best possible Paf1C variant for our studies, but think that in the future ways must be found to use full-length Paf1C when at all possible.

A reference to Fig S1 on Line 176 would help.

We have added a reference to Fig. S1 on Line 176.

3. Supplementary Fig. 3 seems like a logical extension of Fig. 2, and could be combined into the same figure. The legend in this figure should include more details about the conditions of image acquisition e.g. K2 camera, magnification, dose. It would also be good to include representative negative stain micrographs, 2D classes and a 3D reconstruction here, or as a separate figure.

We have added a negative-stain micrograph to Supplementary Fig. 3, and a 3D reconstruction of Pol II-Paf1C-TFIIS EC in Fig. 2, to address this point. We now have included more details about the conditions of image acquisition to the Supplementary Fig. 3.

4. The individual lobe structures presented in Supplementary Fig. 4 or 5 should be moved to the main article, in Fig. 3, as these are the main EM results of the paper. The merged “three-lobed” structures shown in Fig. 3, 5 and 7 represent an interpretation of the three separate structures and should be defined as such. Do volumes A, B and C clash or overlap with each other when merged to generate the composite?

We would be happy to move Supplementary Fig. 4 or 5 to the main article, but have not done so due to space restraints. Instead we now have added views of the three different 3D-classes to Fig. 3, and we have clarified the overlap between the Paf1C regions in the text. See also the response to major point 2. This should take care of this point.

5. To present the cross-linking data (Figure 4d), it would be better to show the crystal structure of Pol II with crosslinked residues highlighted. This would allow the reader to identify surface patches that interact with each of the Paf1C subunits. The locations of the three separate cryo-EM lobes could be indicated beside the structure as panels allowing visualization of the relative proximity of the crosslinked Pol II residues to the separate blobs of EM density.

We have included the Supplementary Fig. 6 showing the highlighted crosslinked residues on Pol II-TFIIS EC structure. In this figure, we also summarized the crosslinked lysine residues of Paf1C.

6. The phrase “movie stacks” should be replaced with “micrographs” (line 145).

We have changed the text accordingly.

7. The colors in Fig. 5a should be changed: Both Paf1C density and TFIIF subunits are different shades of pink/purple, which is confusing.

To be consistent with our previous publications, we would like to leave TFIIIF subunits in plum and medium purple. But we rendered the ribbon model of TFIIIF in a different way to make it more visible.

8. The resolutions listed in lines 156 and 160 are misleading, as this only refers to the Pol II core (see Supplementary Fig. 5e-g). The actual resolutions for densities A, B and C are more like 10 – 20 Å. This should be clearly stated in the text.

We now have stated this in the text.

9. Referring to Cdc73, the authors state “we could fit this crystal structure to the globular density” (line 227), but the actual fit is not shown (apart from an outline in Fig. 4d.) A figure should be included to show this fit in 3D.

We have now replaced the outline of Paf1C density with the EM density in Fig.4d. This shows the fit nicely.

10. The statement in the discussion “thus our results elucidate the general transcription cycle” (line 289) should be removed.

We have rephrased this but wish to maintain the take-home message that the structure helps to explain factor exchange on the Pol II surface, i.e. progression through the transcription cycle.

11. Typo (line 724) “starting form”

OK.

12. In the legend for Fig. 3 (line 844) the authors state “The same threshold level in Chimera was used to contour densities”. It is unclear to what densities this refers – A, B, C and Pol II? Or just A, B and C? It looks like Pol II is displayed at a different threshold to the other densities.

We used Pol II as an indication to put the same threshold. Pol II, A, B, and C are at the same threshold. Because the three reconstitutions were locally filtered and re-normalized and Pol II has much better resolution, so it looks at a different threshold than the densities from Paf1C. We have now stated this in the legend of Fig. 3.

13. In Fig. 4d, it would be useful to include an approximate measure of length on the indicated crosslinks.

We now have included distances between crosslink sites on the Cdc73 GTPase-like domain and Rpb3/Rpb11.

Reviewer #2 (Remarks to the Author):

In this study, Xu et al. use cryo-electron microscopy, chemical crosslinking/mass spectrometry, and protein binding studies to investigate the structure of the yeast Paf1C bound to Pol II and TFIIS. Paf1C has been implicated in regulating chromatin structure during transcription elongation but little is known about this conserved complex at the structural level. Through their cryo-EM analysis, the authors provide the first structural insights into the Paf1 complex and its interactions with Pol II. This is an important contribution to the transcription/chromatin field, as previous crystallography studies focused on small functional domains in Paf1C. Other conclusions from this work indicate that the elongation factor TFIIS can enhance Paf1C association with Pol II and that the initiation factor TFIIF may preclude binding of Paf1C to Pol II through a steric clash. Nascent transcript profiling studies support an important role for Paf1C in global regulation of transcription; however, the impact of this part of the paper is limited by the analysis that was performed. Overall, the primary strength of this paper is the cryo-EM structure of Paf1C and its association with Pol II and TFIIS. The other contributions of the paper are weaker primarily because the data indicate somewhat subtle effects that were analyzed by only a single approach or because the results were not fully analyzed.

We thank the reviewer for the thorough review and highly appreciate the comments and suggestions. We now have included the statistical analysis of Pol II-Paf1C-TFIIS binding assays in Supplementary Fig.2, a native PAGE analysis in Fig.5, and further analysis of 4tU-seq in Fig.6 to strengthen our conclusion.

1. The introduction is not very well written. It reads like a list of facts with insufficient cohesion.

It is true we tried to be as complete as possible in surveying the literature on Paf1 and this leaves the impression of a facts list. We made an effort to group different findings into paragraphs with a common topic, for example chromatin functions of Paf1C. We prefer being balanced and complete in our review of the available literature over a smoother text flow and trust the reviewer agrees.

2. The supplemental figures are cited out of order. The figures should be re-formatted accordingly.

We now have changed the order and corrected this.

3. There is insufficient information on how the limited proteolysis data were analyzed to define the core Paf1C.

We now have explained this in the text. The original idea for the various truncations was to remove flexible regions for crystallization trials. First we purified the full-length Paf1C. Then the limited proteolysis coupled to Edman sequencing showed the C-terminal region of Ctr9 is not stable. We prepared the Paf1C-Ctr9-Δ913 complex. Comparing to the full-length, it expresses better and has less degradation. The limited proteolysis also showed the unstable Leo1 and Paf1, we used iterative truncation strategy, and prepared the core Paf1C.

4. The term “synergistic” seems inappropriate when describing enhanced binding of Paf1C in the presence of TFIIS. Does Paf1C enhance the binding of TFIIS?

We now have modified the text to ‘TFIIS enhances Pol II-Paf1C binding’.

5. Supplementary Figure 2C: The authors report enhanced binding of Paf1C derivatives in the presence of TFIIS. This is very difficult to see in the figures provided. The bands in this and other gels are faint, making comparisons very difficult. Putting the overall faintness of the gel aside, I do not see a significant difference in Paf1C subunit levels between reactions containing TFIIS and those lacking TFIIS. Ctr9 and Cdc73 levels look nearly identical among the four samples in lanes 6-9. Moreover, coomassie staining is not a very quantitative measure of protein abundance.

To make this clearer, we quantified the bands using Image J to visualize the change. We now added quantified Leo1/Rpb4 fold change from 3 replicates in Supplementary Fig.2. In Fig. S2d, error bars indicate the standard error of the mean from 3 replicates.

6. The assignment of specific subunits to the three Paf1C parts (A, B, and C) was guided by chemical crosslinking data. Do the authors have any other information to verify the assignments? For example, Ctr9 is assigned to part B because it crosslinks to Paf1/Leo1 (assigned to A) and Cdc73 (assigned to B). This seems logical but the same argument could be made for Rtf1, which is not assigned to any part of the structure.

We assigned Ctr9 to part B because it crosslinks to Paf1/Leo1 and Cdc73, and is located in between those (Fig. 4). Rtf1 doesn’t crosslink to Cdc73, thus it is difficult to assign it to density B. In our old Fig. 4 right panel, it was not clearly shown that Cdc73 doesn’t crosslink to Rtf1. We have now modified Fig. 4 right panel to show the crosslinks between Paf1C subunits. Since the C-terminal small region of Rtf1 crosslinks to Ctr9, Leo1, and Paf1, we then included this interaction in our model (Fig. 7).

7. Can the authors fit the Paf1-Leo1 crystal structure into Paf1C part A?

We have tried to fit the human Paf1-Leo1 crystal structure into part A density. This crystal structure is only a small part of the Paf1-Leo1 heterodimer (183 aa out of 1197 aa) and does not fit well into the much larger part A density. Thus, yes, we tried, but the fitting was not conclusive.

8. Qiu et al. reported that Cdc73 binds to the CTD of Rpb1. The authors do not address this interaction in their structure or in their description of the model.

The CTD is a long and tethered to the Pol II core with a flexible linker. The region 201-229 aa of Cdc73 is crucial for CTD binding in that publication, which is not included in the crystal structure of GTPase-like domain. Thus we can not pinpoint where the interaction takes place. We haven’t included this interaction in our model, but we have added a sentence to the text to discuss it.

9. The conclusion that binding of TFIIF to Pol II impedes the binding of Paf1C due to a steric clash is intriguing and supports the notion that transition from initiation to elongation is controlled by an exchange of Pol II-bound factors. However, the data for this are not especially strong. In panels 5B and 5C, binding of Paf1C and TFIIF do not appear to be mutually exclusive. Moreover, the staining is faint and not quantitative. In the absence of additional biochemical or functional data, the conclusion of a competition between TFIIF and Paf1C is not yet convincing. A more minor point: the figures could be made clearer by using colors other than shades of purple/pink for both Paf1C and TFIIF.

We have included a native PAGE gel analyzing fraction of the sucrose gradient experiment, showing that TFIIF binds more strongly to Pol II under our conditions. TFIIF can almost completely compete out the Pol II-bound Paf1C. We also agree with the reviewer that the binding may not be mutually exclusive, and we have modified the text accordingly. To be consistent with our previous publications, we would like to leave TFIIF subunits in plum and medium purple. But we increased the ribbon width for TFIIF to improve its visibility.

10. The 4tU-seq data do not add very much to the paper. They are under-analyzed and appear to be tacked on. I would recommend omitting the data or providing a more complete analysis. Also, do the authors have an explanation for why the *rtf1* mutant would show a greater decrease in mRNA levels than the *paf1* mutant? For many phenotypes, *paf1* mutants are more severely compromised than *rtf1* mutants.

*We now have included the analysis of significantly repressed transcripts in $\Delta Paf1$ and $\Delta Rtf1$ strains with the use of a Venn diagram. This analysis shows most repressed transcripts in the $\Delta Paf1$ strain are affected in the $\Delta Rtf1$ strain, but the $\Delta Rtf1$ strain has also 35% uniquely repressed genes, supporting the idea that *Rtf1* has independent roles in transcription elongation (Cao. et. al, 2015, Mol. Cell. Biol. doi:10.1128/MCB.00601-15).*

We further analyzed the types of transcripts repressed in $\Delta Paf1$ and $\Delta Rtf1$ strains. In both cases, the decreases of different types of transcripts shows a similar pattern.

(The repressed types of transcripts in $\Delta Paf1$ and $\Delta Rtf1$ strains. ORF-T: Open Reading

Frame Transcripts; CUT/XUT: Cryptic Unstable Transcripts/ Xrn1-dependent Unstable Transcripts; SUT: Stable Uncharacterized Transcripts; snRNAs: Small Nuclear RNAs.) Additionally, we analyzed the significantly repressed ORF-T transcripts in Δ Paf1 and Δ Rtf1 strains with a Venn diagram (compare with Fig. 6b in the main text). We got similar results compared with the analysis of all types of transcripts shown in Fig.6b.

Repressed ORF-T transcripts in Δ Paf1 and Δ Rtf1 strains by Venn diagram

The Δ Rtf1 mutant has an additional 1,107 repressed transcripts compared with Δ Paf1 mutant. One explanation could be that the repression of these RNAs might compensate or affect cell growth.

11. How was the spike-in normalization carried out? More details would be helpful.

We now have added more details to explain this.

12. Line 58: “Paf1C” should replace “Paf1”

OK.

13. Lines 203-206: The finding of a crosslink between Leo1 and TFIIS is new and interesting. However, with the current writing, the authors suggest it provides a “positive control”.

We have changed the text to make this more appealing.

14. Line 214: What is the evidence for a “subassembly” involving Paf1-Leo1-Ctr9?

The crosslinking results show the proximity of Ctr9 and the Paf1-Leo1 dimer. Additionally, we expressed and purified the Ctr9-Paf1-Leo1 trimer. We now included this into Supplementary Fig. 1.

15. Figure 1: The black underlines are difficult to see. The diagram of Cdc73 should be corrected to read “GTPase-like domain”.

We now have modified this figure accordingly.

16. Figure 2A: What do the curved lines at the bottom of the marker lane represent?

These curved lines were from a bubble during the gel scanning. To avoid the confusion, we now have removed this lane.

17. Supplementary Figure 1a legend: “Crt9” should be changed to “Ctr9”.

OK.

18. Abstract: Although Paf1C is clearly important for mRNA transcription, the term “required for mRNA transcription” is a bit strong. Yeast cells lacking the complex are viable.

We do not wish to substantially change this, but we reworded “required for mRNA transcription” to “required for normal mRNA transcription”. We note that the advantage of 4tU-seq is that one can really see effects on RNA synthesis, i.e. transcription, directly. As it is possible that a factor is required for normal transcription but its deletion does not result in lethality, the statement is fine. Please note we showed earlier that cells that are impaired in transcription can compensate this by increasing mRNA half life, a phenomenon we called mRNA level buffering. We trust the reviewer agrees we can leave this as is.

Reviewers' Comments:

Reviewer #1 (Remarks to the Author):

The authors have done an excellent job of addressing my comments and clarifying several aspects of the work. I have no remaining concerns.

Reviewer #2 (Remarks to the Author):

This very well done study by Xu et al. provides new insights on the physical association of Paf1C with a Pol II elongation complex (EC) containing TFIIS. The manuscript has been improved on multiple fronts: by inclusion of new data (native gels and quantitation), which support inhibition of Paf1C binding to the Pol II EC by TFIIF and stimulation of Paf1C binding to the Pol II EC by TFIIS, by the presentation of additional EM results, and by clearer descriptions of the methods that led to the structural analysis and modeling. Overall, this is an excellent study that lends new insights into the structure and Pol II-association of an important transcription factor complex, which broadly affects gene expression and chromatin structure. Prior to publication, some minor revisions should be made:

1. Figure 1. An additional structurally resolved region in yeast Rtf1 was recently reported by Van Oss et al. (Mol Cell 64: 815) and should be highlighted in panel A.
2. Figure 3A and B legend. Do the authors mean “restitutions” or “reconstructions”?
3. Figure 6, panel A and legend. Correct nomenclature of gene deletions is *paf1Δ* and *rtf1Δ* not Δ Paf1 and Δ Rtf1.
4. The quantitation of Leo1 levels relative to Rpb4 in Figure S2 is an important addition as it solidifies the conclusion that TFIIS stimulates Paf1C binding to Pol II. With that said, the new panel (Fig. S2d) is not well described. Are the data normalized to binding reactions lacking TFIIS? Can “no change” be better defined? An alternative presentation, which might be clearer for the reader, could include the –TFIIS samples and also include more informative labels in the bar graph. Similarly, the description of the quantitation in the legend and on page 16 under “Pull-down assay” is difficult to understand as written. Revisions to the writing are needed.

Detailed list of responses to reviewers' concerns

NCOMMS-16-27989-A

Xu et al. "Architecture of the RNA polymerase II-Paf1C-TFIIS transcription elongation complex"

Reviewers' comments are shown in regular font and our comments in *italics*.

We were pleased that the reviewers are satisfied with the way we addressed their comments and we appreciated the opportunity revise our manuscript. We would like to thank the reviewers for the review process and are very happy that they now all recommend the manuscript for publication.

REVIEWERS' COMMENTS:

Reviewer #1 (Remarks to the Author):

The authors have done an excellent job of addressing my comments and clarifying several aspects of the work. I have no remaining concerns.

Reviewer #2 (Remarks to the Author):

This very well done study by Xu et al. provides new insights on the physical association of Paf1C with a Pol II elongation complex (EC) containing TFIIS. The manuscript has been improved on multiple fronts: by inclusion of new data (native gels and quantitation), which support inhibition of Paf1C binding to the Pol II EC by TFIIF and stimulation of Paf1C binding to the Pol II EC by TFIIS, by the presentation of additional EM results, and by clearer descriptions of the methods that led to the structural analysis and modeling. Overall, this is an excellent study that lends new insights into the structure and Pol II-association of an important transcription factor complex, which broadly affects gene expression and chromatin structure. Prior to publication, some minor revisions should be made:

1. Figure 1. An additional structurally resolved region in yeast Rtf1 was recently reported by Van Oss et al. (Mol Cell 64: 815) and should be highlighted in panel A.

We have added this recently published structure information in our Figure 1, panel a.

2. Figure 3A and B legend. Do the authors mean “restitutions” or “reconstructions”?

We have changed “restitutions” to “reconstructions” in Figure 3a and b legend.

3. Figure 6, panel A and legend. Correct nomenclature of gene deletions is *paf1Δ* and *rtf1Δ* not Δ *Paf1* and Δ *Rtf1*.

*We have changed Δ *Paf1* and Δ *Rtf1* to *paf1Δ* and *rtf1Δ* accordingly in Figure 6 and legend.*

4. The quantitation of *Leo1* levels relative to *Rpb4* in Figure S2 is an important addition as it solidifies the conclusion that TFIIS stimulates *Paf1C* binding to Pol II. With that said, the new panel (Fig. S2d) is not well described. Are the data normalized to binding reactions lacking TFIIS? Can “no change” be better defined? An alternative presentation, which might be clearer for the reader, could include the –TFIIS samples and also include more informative labels in the bar graph. Similarly, the description of the quantitation in the legend and on page 16 under “Pull-down assay” is difficult to understand as written. Revisions to the writing are needed.

The fold change is set to 1 in cases lacking TFIIS (representing unchanged amounts). We have revised the description in the Supplementary Figure 2d legend and also the Methods. We didn't include the TFIIS lacking binding reactions because it is set to 1, but we has revised Supplementary Figure 2d to a clearer view.